


# Observation-Constrained Kinetic Modelling of Isoprene SOA Formation in the Atmosphere

Chuanyang Shen[1], Xiaoyan Yang[2], Joel Thornton[3], John Shilling[4], Chenyang Bi[5], Gabriel Isaacman-VanWertz[6], Haofei Zhang[1]*

[1]Department of Chemistry, University of California, Riverside, CA, 92507, USA
[2]Department of Environmental Sciences, University of California, Riverside, CA, USA
[3]Department of Atmospheric Sciences, University of Washington, Seattle, Washington, USA
[4]Atmospheric, Climate, and Earth Sciences Division, Pacific Northwest National Laboratory, Richland, WA, USA
[5]Center for Aerosol and Cloud Chemistry, Aerodyne Research Inc., MA, USA
[6]Department of Civil and Environmental Engineering, Virginia Tech, Blacksburg, Virginia, USA

*Correspondence to*: Haofei Zhang (haofei.zhang@ucr.edu)

**Abstract.** Isoprene has the largest global non-methane hydrocarbon emission, and the oxidation of isoprene plays a crucial role in the formation of secondary organic aerosols (SOA). Two primary processes are known to contribute to SOA formation from isoprene oxidation: (1) the reactive uptake of isoprene-derived epoxides on acidic or aqueous particle surfaces and (2) the absorptive gas-particle partitioning of low-volatility oxidation products. In this study, we developed a new multiphase condensed isoprene oxidation mechanism that include these processes with key molecular intermediates and products. The new mechanism was applied to simulate isoprene gas-phase oxidation products and SOA formation from previously published chamber experiments under a variety of conditions and atmospheric observations during the Southern Oxidant and Aerosol Studies (SOAS) field campaign. Our results show that SOA formation from most of the chamber experiments is reasonably reproduced using our mechanism except when the concentration ratios of initial nitric oxide to isoprene exceeds ~2. The SOAS simulations also reasonably agree with the measurements regarding the diurnal pattern and concentrations of different product categories. The molecular compositions of the modelled SOA indicate that multifunctional low-volatility products contribute to isoprene SOA more significantly than previously thought, with a median mass contribution of ~57% to the total modelled isoprene SOA. This contribution, however, may vary greatly, mainly dependent on the volatility estimation and treatment of particle-phase processes (i.e., photolysis and hydrolysis). Our findings emphasize that the various pathways to produce these low-volatility species should be considered in models to more accurately predict isoprene SOA formation. The new condensed isoprene chemical mechanism can be further incorporated into regional-scale air quality models, such as the Community Multiscale Air Quality Modelling System (CMAQ), to assess isoprene SOA formation on a larger scale.

## 1 Introduction

Isoprene (2-methyl-1,3-butadiene, $C_5H_8$) is a highly reactive hydrocarbon that is widely recognized as the most abundant biogenic volatile organic compound (BVOC) emitted into the atmosphere (Guenther et al., 2006). Given its large flux globally



and high reactivity, isoprene plays a key role in affecting the balance of atmospheric trace species such as $O_3$, $NO_x$ (= NO + $NO_2$), and $HO_x$ radicals (= OH + $HO_2$), and is also a significant source of secondary organic aerosol (SOA) (Edney et al., 2005;

Kroll and Seinfeld, 2005; Dommen et al., 2006; Henze and Seinfeld, 2006; Kroll et al., 2006; Surratt et al., 2006; Lewandowski et al., 2008; Fu et al., 2009; Jaoui et al., 2010; Surratt et al., 2010). Field measurements indicate high SOA mass concentrations from isoprene, which can reach 4 µg m$^{-3}$ or even higher in summertime (Claeys et al., 2004a; Kourtchev et al., 2005; Budisulistiorini et al., 2013; Hu et al., 2015). Therefore, understanding the isoprene atmospheric oxidation and the corresponding SOA formation mechanisms are of crucial importance for accurate estimation of ambient PM$_{2.5}$ mass

concentrations, especially in BVOC-dominant regions.

The gas-phase oxidation of isoprene initiated by both hydroxyl radicals (OH) and nitrate radicals ($NO_3$) has been extensively investigated in laboratory and computational studies (Mcgivern et al., 2000; Paulot et al., 2009; Peeters and Nguyen and Vereecken, 2009; Crounse et al., 2011; Teng and Crounse and Wennberg, 2017; Wennberg et al., 2018; Berndt et al., 2019; Guo et al., 2020; Vereecken et al., 2021; Zhang et al., 2022; Tsiligiannis et al., 2022), with many different chemical

mechanisms having been proposed (Paulson and Seinfeld, 1992; Carter, 1996; Pöschl et al., 2000; Fan and Zhang, 2004; Taraborrelli et al., 2009; Zhang et al., 2011; Wennberg et al., 2018). Likewise, in the particle phase, many isoprene oxidation-derived molecular species have also been reported in aerosol samples from field and laboratory studies that can provide insight into the SOA formation mechanisms (Claeys et al., 2004b; Kourtchev et al., 2005; Edney et al., 2005; Carlton and Wiedinmyer and Kroll, 2009; Surratt et al., 2010; Lin et al., 2012; Lin et al., 2013; Liu et al., 2016; Schwantes et al., 2019). While great

advances have been made in both the gas and particle phases, a comprehensive molecular-level isoprene SOA model is lacking, partly because of the highly complex processes that can contribute to the isoprene SOA under different conditions. In general, two primary pathways contribute to SOA formation from isoprene oxidation: (1) the reactive uptake of epoxides on acidic or aqueous particle surfaces (Surratt et al., 2010; Lin et al., 2012; Lin et al., 2013; Nguyen et al., 2014; Y. Zhang et al., 2018), and (2) the gas-particle absorptive partitioning of multifunctional low-volatility compounds which are usually formed via

multigenerational oxidation (Krechmer et al., 2015; Liu et al., 2016; Schwantes et al., 2019).

SOA formation from the reactive uptake of epoxides generally refers to the ring-opening reactions of isoprene-derived epoxydiols (IEPOX) onto aerosols catalyzed by acidity or water. The main SOA products from this pathway include the 2-methyltetrols (2-MT), C5-alkenetriols, and isoprene-derived organosulfates (IEPOX-OS) (Surratt et al., 2010; Lin et al., 2012). Both SOA yield and composition from this pathway vary greatly and depend on many factors, such as particle surface area,

particle acidity, and particle phase state (Gaston et al., 2014; Nguyen et al., 2014; Xu et al., 2015; Y. Zhang et al., 2018; Yee et al., 2020). Owing to its reported substantial contribution to ambient PM$_{2.5}$, this pathway has been extensively studied in prior laboratory research (Lin et al., 2012; Nguyen et al., 2014). Many model studies have also attempted to explicitly model SOA production from IEPOX reactive uptake (Pye et al., 2013; Budisulistiorini et al., 2015). In addition to IEPOX, other epoxide and lactone species, such as methacrylic acid epoxide (MAE) and hydroxymethel-methyl-α-lactone (HMML), have

also been suggested as significant contributors to isoprene SOA in the presence of $NO_x$ (Lin et al., 2013; Riedel et al., 2015; Nguyen et al., 2015b). Meanwhile, low-volatility products from multigenerational oxidation may also contribute to isoprene





SOA, especially for those that maintain the five-carbon moiety (C5-LV). A well-studied example is that isoprene hydroxy hydroperoxide (ISOPOOH), the direct precursor of IEPOX, was found to undergo OH-oxidation and form highly oxidized low-volatility products that form SOA (see Fig. 1) (Krechmer et al., 2015; Liu et al., 2016; D'ambro et al., 2017; Mettke et al.,

2023). In addition, N-containing multifunctional low-volatility species (C5-NLV) from chamber and field measurements suggests that further oxidation of isoprene nitrates could be another SOA source (Lee et al., 2016; Schwantes et al., 2019). These low-volatility isoprene nitrates could be formed from $NO_x$-involved pathways during OH oxidation or during nighttime $NO_3$ oxidation (Fig. 1) (Ng et al., 2008; Schwantes et al., 2015; Schwantes et al., 2019).

**Figure 1. Simplified reaction scheme for isoprene oxidation by OH and NO₃. The major low-volatility species that may contribute to SOA formation are highlighted in dashed boxes. For simplicity, RO₂ + RO₂ reactions and products are not shown.**



While the basic understanding of these pathways and their contribution to isoprene SOA formation have been established, they have not been fully incorporated into chemical models especially for ambient SOA simulations. Instead, most regional and

global models use highly simplified gas-phase isoprene oxidation condensed mechanism such as the Carbon Bond mechanisms (Gery et al., 1989; Yarwood and Whitten and Rao, 2005; Yarwood et al., 2010) and the SAPRC mechanisms (Carter, 2000; Carter, 2010; Carter, 2023). Most of these mechanisms are too condensed to comprehensively represent these low-volatility products that are important for SOA formation (Perring et al., 2009; Archibald et al., 2011). Furthermore, the SOA formation in these models use volatility-based yield parameterizations (Odum et al., 1996; Donahue et al., 2006). Because these

parameterizations are derived from laboratory SOA mass concentrations formed under specific oxidation conditions, it may bring large uncertainties to the SOA estimation under realistic conditions (Marais et al., 2016). In order to better interpret and predict isoprene SOA, a more detailed representation of the isoprene gas-phase oxidation processes is needed in chemical mechanisms, especially for the products that are relevant to SOA formation. In a recent study, Thornton et al. (2020) incorporated and modified the near-explicit Master Chemical Mechanism (MCM) into a dimensionless (0-D) model to simulate

SOA formation from chamber studies. Their model estimations agree well with the observations in SOA mass concentrations. However, the MCM-based model is too large to be applied to regional or global models. Moreover, the extensive isomer-resolved details in MCM are unnecessarily all accurate or needed in large-scale models.

Therefore, a multiphase isoprene oxidation mechanism with intermediate level of chemical details is needed. It should be based on a condensed chemical mechanism but expanded for isoprene chemistry to an appropriate extent to include the major SOA

species. It should also be flexible for the inclusion of any new isoprene chemistry that is reported in laboratory, mechanistic and field studies [e.g., (Wennberg et al., 2018; Vasquez et al., 2020; Mettke et al., 2022; Carlsson et al., 2023)]. Lastly, this mechanism should be easily implementable into regional or global air quality models to better represent isoprene chemistry and SOA formation.

In this study, we developed such a new condensed isoprene chemical mechanism adapted to the SAPRC structure (Carter,

2010) and incorporated this mechanism into a box model to simulate existing isoprene oxidation chamber experiments under various initial conditions (e.g., OH oxidation, $NO_3$ oxidation, and different $NO_x$ levels, etc.). The key gas-phase products from all the pathways described above and SOA mass concentrations are compared with laboratory observations (where available) and other chemical mechanisms to evaluate the mechanism's performance. We also applied the new mechanism to model the 2013 Southern Oxidant and Aerosol Studies (SOAS) field campaign at the Centreville, AL site (Lee et al., 2016; H. Zhang et

al., 2018), to elucidate the relative importance of the various pathways in SOA formation under real atmospheric conditions. To the best of our knowledge, this is the first time that a comprehensive molecular-level isoprene SOA mechanism is evaluated using field observations. Lastly, we also discuss the major uncertainties in current mechanistic understandings and the needed future research directions regarding isoprene SOA chemistry.





## 2 Model Descriptions

### 2.1 F0AM-WAM


We use the Framework for 0-dimensional Atmospheric Modelling (F0AM v3.2) (Wolfe et al., 2016) coupled to the Washington Aerosol Module (WAM), denoted as F0AM-WAM, to simulate the isoprene oxidation processes and predict SOA formation and evolution (Thornton et al., 2020). F0AM is a flexible and efficient MATLAB-based framework for modelling 0-dimensional atmospheric chemistry and it allows for easy incorporation of new and modified chemical mechanisms to simulate a variety of typical problems, including photochemical chambers and field observations from ground and aircraft (Brune et al., 2021; Lyu et al., 2022). The WAM is a specialized module designed to simulate the formation and evolution of SOA by explicitly treating the condensation and evaporation of low-volatility compounds. In combination, F0AM-WAM provides a comprehensive tool for studying the interactions between atmospheric gas-phase chemistry and aerosol processes.


### 2.2 Gas-Phase Chemistry


A new isoprene oxidation gas-phase kinetic mechanism was developed in this study, named "UCR-ISOP". It was developed on top of a version of the SAPRC07 mechanism that is currently used in the CMAQ model (i.e., SAPRC07tic) (Carter, 2010; Xie et al., 2013). The other chemical mechanisms discussed in this work include: the MCMv3.3.1 mechanisms (denoted as "MCM" below); the Caltech isoprene mechanism (the "reduced_plus_v5" version, denoted as "Caltech" below) summarized by Wennberg et al. (2018); the modified MCM mechanism by Thornton et al. (2020) (denoted as "MCM-UW" below); and the isoprene mechanism proposed by the Forschungszentrum Jülich institution (denoted as "FZJ" below) (Vereecken et al., 2021; Tsiligiannis et al., 2022; Carlsson et al., 2023). In UCR-ISOP, most of the existing isoprene related reactions in the SAPRC07 mechanism were replaced by new reactions (simplified in Fig. 1) that were condensed in large part from the Caltech isoprene mechanism and the MCM mechanism. For the isoprene + $NO_3$ reactions, the new FZJ mechanism proposed by recent studies was also incorporated to some extent as discussed later (Vereecken et al., 2021; Tsiligiannis et al., 2022; Carlsson et al., 2023). Certain isomers are individually represented (with lumping in some cases) for several major species, including the isoprene hydroxyl peroxy radicals (ISOPOHOO, 2 isomers), hydroperoxyl aldehydes (HPALD, 2 isomers), ISOPOOH (3 isomers), IEPOX (2 isomers), and isoprene hydroxynitrates (IHN, 3 isomers). These species have been extensively studied in the literature and distinct reactivities and reaction products have been reported (Wennberg et al., 2018). Maintaining some of the lumped isomers for these species permits more accurate representations of their further product distributions. Other than these species, the isomers with the same functional groups were lumped as identical compounds. For example, each of the low-volatility species shown in Fig. 1 are described as an individual compound that could represent the sum of several isomers. All the abbreviated names in the UCR-ISOP mechanism are described in Table S1 in the Supplementary Material. The naming convention is the same as in the Caltech mechanism, except for species that were already included in the SAPRC07tic mechanism. Compared to SAPRC07tic (38 species and 124 reactions), UCR-ISOP adds 39 additional species and 118







additional gas-phase reactions. In comparison, the MCM mechanism has 610 species and 1974 reactions (related to isoprene); the Caltech mechanism has 155 species and 429 reactions.

In addition to the above-mentioned features, the new mechanism also includes (1) temperature and pressure dependence of organic nitrate yield from peroxy radical (RO$_2$) + NO reactions as suggested by the Caltech mechanism; (2) isomerization reactions for the major RO$_2$ based on recent studies (D'ambro et al., 2017; Wennberg et al., 2018; Vereecken et al., 2021;

Mettke et al., 2023); and (3) dimer formation from several RO$_2$ + RO$_2$ reactions that were supported by prior chamber experiments (Ng et al., 2008; Mettke et al., 2023). This mechanism was implemented into F0AM-WAM to simulate published chamber experimental data under different conditions. The model outputs are compared with both the available measurements and other existing mechanisms (i.e., the Caltech mechanism and the MCM mechanism). It should be noted that the mechanism is highly condensed and simplified for the potential application in regional or global models. Thus, it does not capture all the

known chemical reactions in isoprene oxidation.

**2.3 Gas-Particle Partitioning**

In this work, partitioning of low-volatility or semi-volatility species into the particle phase is parameterized to include two separate processes: absorptive equilibrium partitioning into an organic phase (Pankow, 1994; Odum et al., 1996) and aqueous

uptake by liquid water (Wania et al., 2015; Isaacman-Vanwertz et al., 2016). In general, the condensation kinetics to particle is calculated as:

$$K_{cond} = K_{mt} \times \text{SA}, \tag{1}$$

where $K_{mt}$ is the mass transfer rate (cm s$^{-1}$) and SA is the aerosol surface area per volume (cm$^2$ cm$^{-3}$). The evaporation back to gas phase will be calculated as:

$$K_{evap} = K_{mt} \times \text{SA} \times \left( H_{aq} \times \text{LWC} \times \frac{RT}{10^{12}} + \frac{C_{OA}}{C^*} \right)^{-1}, \tag{2}$$

where $H_{aq}$ is the estimated Henry's law constant (M atm$^{-1}$), LWC is the aerosol liquid water content (μg m$^{-3}$), R is the ideal gas constant, T is the temperature (K), $C_{OA}$ is the organic aerosol mass concentration (μg m$^{-3}$) and C* is the saturation concentration (μg m$^{-3}$). The C* value is calculated from the vapor pressure, which can be estimated using EVAPORATION (Compernolle and Ceulemans and Müller, 2011) and SIMPOL.1 (Pankow and Asher, 2008). The calculated C* for the major

low-volatility and semi-volatility species are listed in Table S2 in the Supplementary Material. When different C* values are estimated for different isomers of each low-volatility species, we used the lowest C*; and the uncertainty of this treatment is tested and discussed later. When the gas-particle equilibrium is established, the fraction of species in the particle phase $F_p$ will be estimated as:

$$F_p = 1 - (1 + H_{aq} * \text{LWC} * \frac{RT}{10^{12}} + \frac{C_{OA}}{C^*})^{-1}, \tag{3}$$

Under dry conditions, $F_p$ will be simplified into 1-(1+$C_{OA}$/C*)$^{-1}$ given that LWC = 0 μg m$^{-3}$.



## 2.4 Reactive Uptake

All the chamber experiments used in this work to evaluate the isoprene multiphase mechanism were performed in the absence of aqueous/acidic sulfate seed aerosols (Kroll et al., 2006; Ng et al., 2008; Schwantes et al., 2015; Liu et al., 2016; D'ambro et al., 2017; Shilling et al., 2019). Therefore, the reactive uptake of IEPOX (and other epoxides) is not expected to occur in these chamber experiments. However, in the SOAS field campaign where aqueous particles containing sulfate were ubiquitous, IEPOX-derived SOA was reported to be an important contributor to total organic aerosol (Hu et al., 2015; Xu et al., 2015; H. Zhang et al., 2018). Therefore, in the application of the model to the field measurements, in addition to the absorptive partitioning of semi-volatility and low-volatility oxidation products, we also consider the SOA formation from IEPOX reactive uptake onto acidic/aqueous aerosols. For modelling simplicity, only 2-MT and IEPOX-OS are assumed to be formed from this process. Therefore, the "2-MT" in the model is likely a summation of 2-MT, C5-alkenetriols, and other minor IEPOX-derived non-OS species. Measurements report C5-alkenetriols as a tracer for IEPOX-derived SOA (Lin et al., 2012), but no formation pathway is known for these compounds. Some work has indicated that they may partly be analytical products of other tracers such as IEPOX-OS (Rattanavaraha et al., 2016; Cui et al., 2018), but their origin remains highly uncertain, so no formation mechanism is included in the mechanism examined here. The IEPOX-SOA formation is parameterized in the model as the following:

$$IEPOX_{(g)} \rightarrow IEPOX\text{-}SOA_{(aerosol)}, \tag{4}$$

$$k_{het} = \frac{SA \times \omega \times \gamma}{4}, \tag{5}$$

where $k_{het}$ is the heterogeneous reaction rate of IEPOX (s⁻¹), SA is the surface area of the aerosol that IEPOX is uptaken onto (cm² cm⁻³), $\omega$ is the mean molecular speed of IEPOX in the gas phase (cm s⁻¹), and $\gamma$ is the reactive uptake coefficient, which can be parameterized using a resistor model from previous studies (Gaston et al., 2014). This resistor model can be calculated as:

$$\frac{1}{\gamma} = \frac{r_p \times \omega}{4 \times D_{gas}} + \frac{1}{\alpha} + \frac{1}{\Gamma_{aq}}, \tag{6}$$

where α is the unitless accommodation coefficient (0.02), $r_p$ is the aerosol particle's radius (cm), $D_{gas}$ is the gas-phase diffusion of IEPOX (cm² s⁻¹), the aqueous term, $\Gamma_{aq}$, is calculated from the following equation:

$$\Gamma_{aq} = \frac{4 \, V R T H_{aq} k_{aq}}{SA * \omega}, \tag{7}$$

where V is the particle volume concentration (cm³ cm⁻³); R is the ideal gas constant; T is the ambient temperature (K); $\omega$ is the gas-phase mean molecular speed (cm s⁻¹) of IEPOX; $k_{aq}$ is the pseudo-first-order rate constant (s⁻¹) defined in Text S1 in the Supplementary Material. The parameter with the largest uncertainty regarding IEPOX reactive uptake is $H_{aq}$, for which prior studies have used values ranging from $1.9 \times 10^7 – 4 \times 10^8$ M atm⁻¹ (Chan et al., 2010; Gaston et al., 2014; Schmedding et al., 2020). Here, we choose to use a median $H_{aq}$ value of $1.3 \times 10^8$ M atm⁻¹, which was predicted by Eddingsaas and Vandervelde





and Wennberg (2010). In model sensitivity analysis, it turns out that the IEPOX-SOA concentration is not very sensitive to the chosen $H_{aq}$ value in this range.

Additional description of this process can be found in Text S1 in the Supplementary Material. In the model, the aerosols were assumed as a homogeneous mixture of organic and inorganic constituents. The LWC is predicted using the thermodynamic

equilibrium model ISORROPIA II based on the measured concentrations of inorganic species, including ammonium, nitrate, and sulfate (Fountoukis and Nenes, 2007). The IEPOX concentration is from the output of the gas-phase oxidation reactions. In addition to IEPOX, several other products from isoprene oxidation may also undergo reactive uptake, such as HMML (Nguyen et al., 2015b), MAE (Lin et al., 2013), 1,2-IHN (Vasquez et al., 2020), glyoxal (Kroll et al., 2005; Carlton et al., 2007) and the other epoxide products included in the mechanism (several examples shown in Fig. 1). The reactive uptake of

these species is included in the model. For HMML and MAE, the major reactive uptake products are 2-methylglyceric acid (MGA) and its corresponding organosulfate. The other epoxides which have not been studied in prior research are assumed to undergo a similar process as IEPOX in the model that form ring-open alcohols and organosulfates. In the case of 1,2-IHN, the reactive uptake product is expected to be a diol (IDH) via hydrolysis that is expected to quickly evaporate back to the gas phase. The reaction rate is calculated from LWC, $H_{aq}$, and the aqueous hydrolysis rate used in Vasquez et al. (2020). For the

reactive uptake of glyoxal, because the equilibrium state is quickly established between aqueous and hydrated glyoxal and the hydrated state is strongly favored, we adopt the $H_{aq}$ of $2.6 \times 10^7$ M atm$^{-1}$ to calculate the glyoxal-derived aqueous SOA (Hastings et al., 2005).

## 2.5 Particle-Phase Reactions

After low-volatility compounds partition to the particle phase, they likely continue to undergo chemical evolution processes. These processes can either decrease organic aerosol mass such as particle-phase photolysis and hydrolysis (Pye et al., 2015; Krapf et al., 2016; Zawadowicz et al., 2020) or promote SOA formation like accretion reactions (Kroll and Seinfeld, 2008). Thus, it is necessary to include or parameterize these particle-phase reactions in models in order to better predict SOA's evolving mass concentration and chemical composition.

In prior work, Surratt et al. (2006) reported substantial formation of peroxides in isoprene SOA formed under NO$_x$-free conditions, which exhibited a pronounced decrease with extended radiation time. Consistently, organic peroxides are believed to be susceptible to photolysis (Chacon-Madrid and Henry and Donahue, 2013) with lifetimes of about 6 days in Los Angeles. Zawadowicz et al. (2020) found that the SOA produced from isoprene oxidation under low-NO$_x$ conditions underwent photolysis-induced mass loss at rates between 1.5–2.2% of NO$_2$ photolysis ($j_{NO_2}$). In order to simulate the SOA decay in the

NO$_x$-free chamber experiments (Kroll et al., 2006; Liu et al., 2016), we apply a first-order photolysis rate coefficient that is 2% of $j_{NO_2}$ to all the products with one or more hydroperoxide (–OOH) groups formed under NO$_x$-free conditions, as proposed by Thornton et al. (2020). However, under high-NO$_x$ conditions, prior studies did not observe such a rapid SOA mass decay





(Kroll et al., 2006; Schwantes et al., 2019). Thus, we assume that particle-phase organic nitrate products photolyze at similar rate coefficients as for the known gas-phase alkylnitrates, which is much slower.

In the SOAS campaign simulations, because of the high relative humidity (RH) and LWC at the field site, we also apply hydrolysis reactions for the organic nitrate species in the simulated isoprene SOA. We assume that their average lifetime against hydrolysis is 3 hours, through which the $-ONO_2$ group is converted to the $-OH$ group (Pye et al., 2015). Other particle-phase processes such accretion and heterogeneous OH oxidation are not included in the current model because the detailed kinetics and mechanisms are highly uncertain.

## 240   3. Results and Discussions

### 3.1 Simulations of Chamber Isoprene Oxidation Experiments

#### 3.1.1 Description of Chamber Experiments

Isoprene oxidation chamber experiments from previously published datasets were used here to test UCR-ISOP's performance in simulating trace gas species and SOA formation under different conditions. These experiments were designated as UNC-
2010/2012 (Zhang et al., 2011; Zhang et al., 2013), Kroll-2006 (Kroll et al., 2006), PNNL-2014 (Liu et al., 2016), PNNL-2018 (Zawadowicz et al., 2020; Thornton et al., 2020), Schwantes-2019 (Schwantes et al., 2019), Ng-2008 (Ng et al., 2008), Carlsson-2023 (Carlsson et al., 2023), and Schwantes-2015 (Schwantes et al., 2015). Kroll-2006 (Run 1-9) and PNNL-2018 were performed under $NO_x$-free conditions; UNC-2010/2012, Kroll-2006 (Run 9-14), PNNL-2014, and Schwantes-2019 experiments were performed under high-$NO_x$ conditions; Ng-2008, Carlsson-2023 and Schwantes-2015 experiments were
performed under $NO_3$ oxidation conditions. Some of these experiments were only used for gas-phase mechanism evaluation due to unavailable particle measurements and some were also used to evaluate the SOA simulations. The conditions for these chamber experiments can be found in Table S3-7 and additional details can be found in the corresponding literature.

#### 3.1.2 Gas-Phase Modelling

To evaluate the new UCR-ISOP gas-phase mechanism, we compared its simulations against the above-described chamber
experimental data and other existing mechanisms. The gas-phase species' concentrations that are typically available for model evaluations are $O_3$, $NO_x$, and isoprene. Quantitative organic product concentrations, although crucial, are usually unavailable. Therefore, we evaluated the UCR-ISOP mechanism against isoprene-$O_3$-$NO_x$ measurements from prior high-$NO_x$ (UNC-2010/2012) and low-$NO_x$ chamber experiments (Kroll-2006) and against simulated oxidation products between different mechanisms, including the MCM mechanism (Jenkin and Young and Rickard, 2015), the Caltech mechanism (Wennberg et
al., 2018), and the SAPRC07tic mechanism (Carter, 2010; Xie et al., 2013).

Figure 2A–C show representative examples of the simulation performance of the various gas-phase mechanisms for three UNC-2010/2012 high-$NO_x$ experiments with varied initial $NO_x$/isoprene concentration ratios. The results indicate that UCR-





ISOP can reasonably predict the temporal evolutions of isoprene, $O_3$ and NO concentrations under a wide range of initial conditions. The statistical evaluation for all the simulated experiments is summarized in Figure 2D–F using the normalized

root mean square error (NRMSE) as the metric. NRMSE can provide a harmonized assessment of the average magnitude of errors between the predicted and the observed values. It appears that the isoprene and NO decay is modelled very well by UCR-ISOP compared to other mechanisms with a median NRMSE of 0.07 and 0.12, respectively. The $O_3$ concentrations are modelled reasonably well by all four mechanisms with median errors less than 0.15. For the NO prediction, MCM and Caltech mechanisms predicted relatively larger bias against measurements with median errors around 0.2. The performances of all the

compared mechanisms are in general similar to each other. The simulation-measurement comparison for $NO_2$ was not examined because the measured $NO_2$ is interfered by other $NO_y$ species (e.g., $NO_3$, $HNO_3$, alkyl nitrates, $N_2O_5$, etc.) (Zhang et al., 2011).

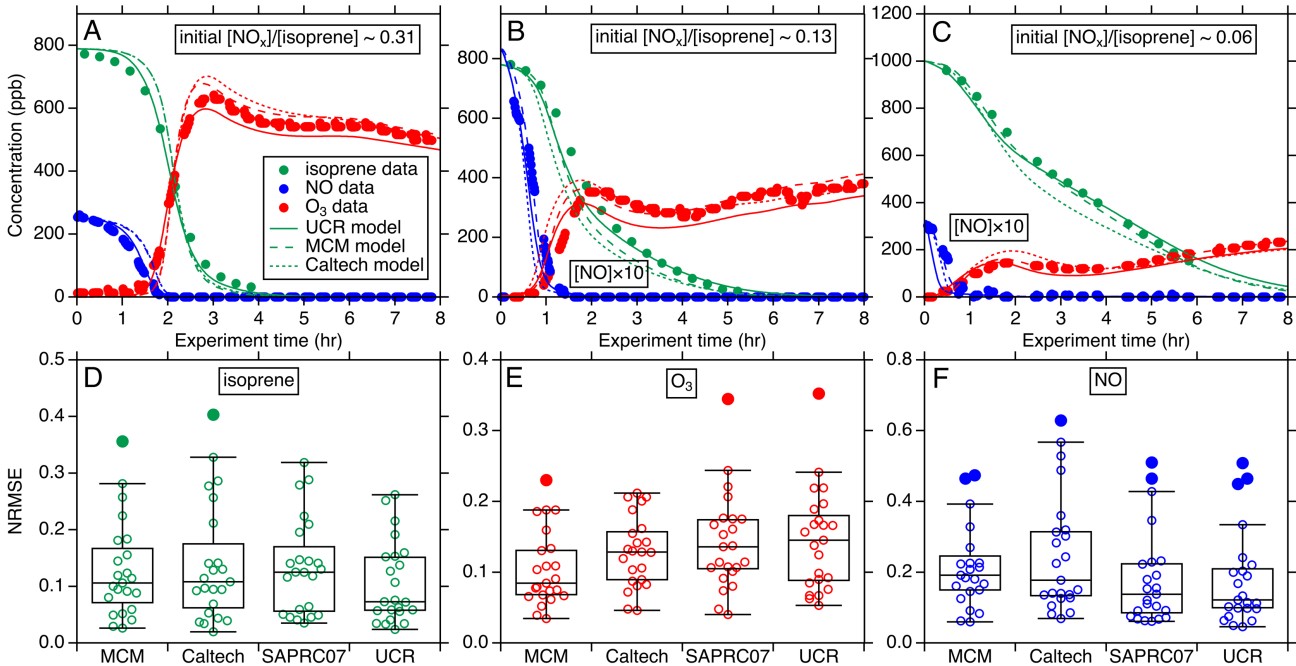

**Figure 2. Intercomparison of simulated isoprene, $O_3$ and NO using different chemical mechanisms (MCM, Caltech, and UCR-ISOP,**
**denoted as "UCR") for the UNC-2010/2012 experiments: (A) 20101021N with an initial $NO_x$/isoprene ratio of 0.31, (B) 20120603N with an initial $NO_x$/isoprene ratio of 0.13, and (C) 20100904N with an initial $NO_x$/isoprene ratio of 0.06. The 3 plots in the lower panel are the NRMSE for all the simulated UNC-2010/2012 and Kroll-2006 experiments using different chemical mechanisms: (D) is the comparison for isoprene; (E) is for $O_3$ and (F) is for NO. NRMSE is defined as the $RMSE/\bar{Y}_{obs}$. $\bar{Y}_{obs}$ denotes the mean of the observed values. In the $x$-axis, "SAPRC07" denotes the SAPRC07tic mechanism. For each box, the central horizontal line in the box**
**indicates the median, and the bottom and top edges of the box indicate the 25th and 75th percentiles, respectively. The whiskers extend to the most extreme data points not considered outliers, and the outliers are plotted using the solid circle markers (other data points plotted using the open circle markers).**



For the other important gas-phase products such as ISOPOOH, IEPOX, methacrolein (MACR), methyl vinyl ketone (MVK),
IHN, glyoxal, methylglyoxal, and all the multifunctional low-volatility compounds, comparisons were made only among
different mechanisms since real-time and quantitative measurements were not available. The results for the major products
and categories (i.e., C5-LV and C5-NLV) from isoprene OH oxidation can be found in Fig. 3 and additional comparisons for
individual species can be found in Fig. S1 in the Supplementary Material. In these comparisons, the predictions for all these
products are generally consistent between the UCR-ISOP, Caltech, and MCM mechanisms with some species and conditions
more scattered than the others. Specifically, the six major individual and groups of products presented in Fig. 3 show very
good agreement between the three mechanisms especially under the lower concentration ranges, suggesting that UCR-ISOP
does not sacrifice model performance during mechanism reduction and can simulate the most important products very well
under most OH oxidation conditions. However, the mechanism comparisons for some individual species exhibit larger
differences, as illustrated in Fig. S1 For example, the two low-volatility products from ISOPOOH + OH oxidation, IDHPE
and IDHDP exhibit opposite trends when comparing between the UCR-ISOP and Caltech mechanisms. This is because we
adopted a slower isomerization rate coefficient for ISOPOOHOO ($C_5H_{11}O_6$, peroxy radical from ISOPOOH + OH, see Fig. 1)
than that used in the Caltech mechanism (Wennberg et al., 2018) and other models (D'ambro et al., 2017; Thornton et al.,
2020). Recent work by Mettke et al. (2023) suggested that the ISOPOOHOO isomerization is ~ 0.002 $s^{-1}$, rather than the order
of 0.1 $s^{-1}$ reported by D'ambro et al. (2017) and Wennberg et al. (2018). Thus, in the UCR-ISOP mechanism, we chose to use
a rate coefficient 10 times slower than the Caltech mechanism (i.e., 0.01 $s^{-1}$), which is between the two very different suggested
rate coefficients. This change greatly affects the yields of IDHPE vs. IDHDP. In addition, we consider the rapid ISOPOOHOO
self-reaction to form the corresponding carbonyl ($C_5H_{10}O_6$), alcohol ($C_5H_{12}O_6$), and dimer ($C_{10}H_{22}O_{10}$), suggested by Mettke
et al. (2023), with a rate coefficient of $1\times10^{-11}$ $cm^3$ molecule$^{-1}$s$^{-1}$. This dimer formation pathway could partly explain the slightly
lower C5-LV simulation using the UCR-ISOP mechanism under high concentrations (Fig. 3E).





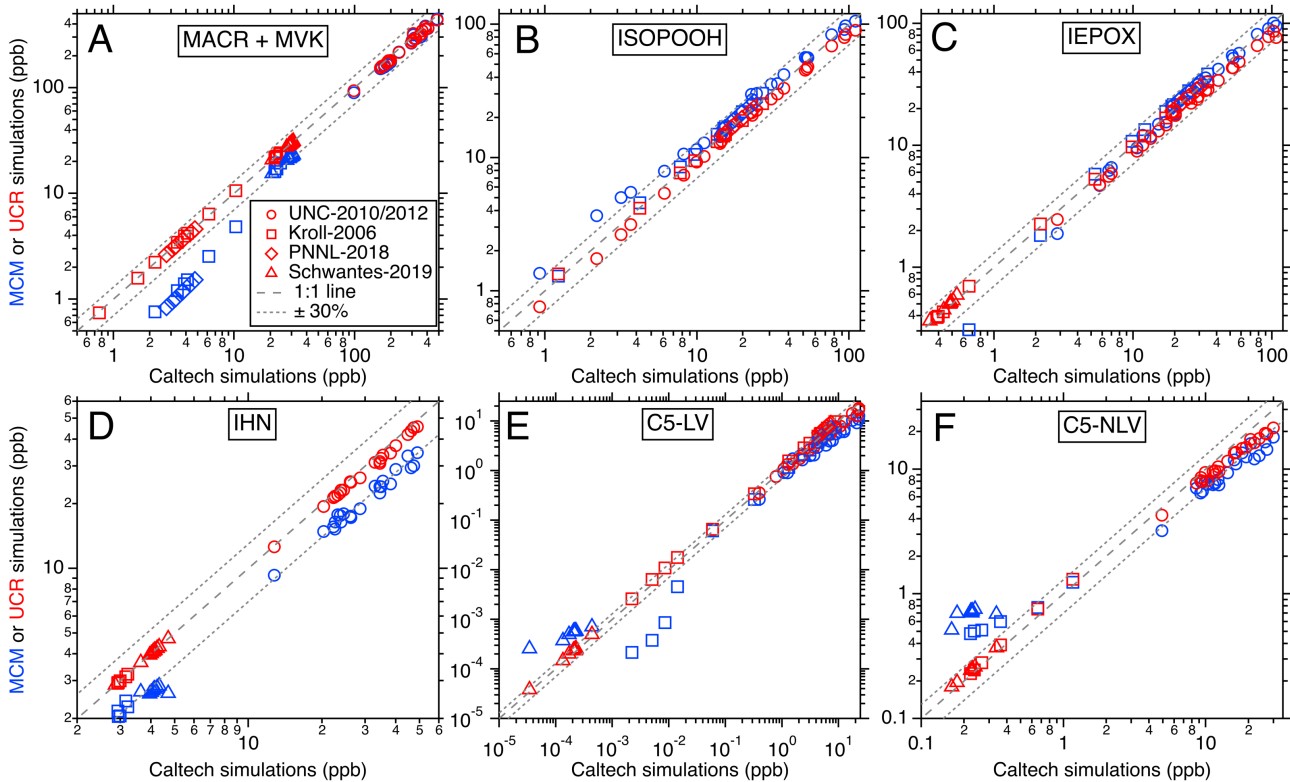

**Figure 3. Comparisons of the simulated isoprene oxidation products' maximum concentrations in each laboratory experiment between different chemical mechanisms. The *x*-axes represent the predictions using the Caltech mechanism and y-axes represent values from UCR-ISOP (red markers) and MCM mechanism (blue markers). Different marker types represent different chamber studies. In all panels, the dashed line indicates a 1:1 correspondence, and the dotted line delineates a 30% uncertainty boundary.**

Moreover, the higher ICPDH predictions in the MCM simulations are due to the fact that the IEPOX-derived $RO_2$ reacting with $HO_2$ is assumed to completely form the ICPDH, while in the Caltech and the UCR-ISOP mechanisms, a much smaller branching ratio (~35%) leads to this hydroperoxide, while the rest leads to RO that subsequently decompose. The latter treatment is likely more reasonable because several isomers of IEPOX-derived $RO_2$ are tertiary $RO_2$ which were suggested to form RO at high branching ratios by reacting with $HO_2$ (Kurtén et al., 2017). Another very different simulated product is glyoxal, for which the UCR-ISOP and the Caltech mechanisms predict lower glyoxal concentrations than the MCM mechanism does by a factor of ~ 3. This result is driven by the combined influence of many reactions. One of the major contributors to glyoxal formation in the MCM mechanism is from RO decomposition of C527O, which stemmed from isomerization of a first-generation RO from isoprene + OH (CISOPCO, see Fig. S2). However, we think this is a less likely pathway considering the multiple complex H shifts involved. Two other major glyoxal contributors in the MCM are from the $NO_3$ oxidation pathway that will be discussed later.

 

The gas-phase isoprene + $NO_3$ mechanism has a large uncertainty and is not consistent among different experimental studies. The key discrepancy lies in the fates of the alkoxy radicals (INO) from the primary nitrate-$RO_2$ (NISOPO2) reacting with $NO_3$, $HO_2$, and $RO_2$. In the Caltech mechanism (Wennberg et al., 2018), the β-isomers of INO exclusively undergo dissociation to

form MACR/MVK, formaldehyde, and $NO_2$, while the δ-isomers of INO mainly isomerize or add $O_2$ to form isoprene carbonyl nitrates (ICN) and $HO_2$. This mechanism was constrained by prior laboratory measurements by Schwantes et al. (2015). In contrast, the recent FZJ mechanism suggests that all the INO isomers predominantly undergo ring-closure reactions to form epoxide-containing $RO_2$. The two different mechanisms could lead to significant discrepancies in the concentrations of MACR/MVK, ICN, $HO_2$, and $NO_2$. Carlsson et al. (2023) suggested that the FZJ mechanism is likely more accurate by

showing that the Caltech mechanism significantly overpredicts MACR+MVK measured in experiments due to INO fragmentation. However, Carlsson et al. (2023) also stated that the FZJ mechanism underpredict $HO_2$, suggesting missing sources. In the UCR-ISOP mechanism, we seek to reach a balance between the two isoprene + $NO_3$ mechanisms. We tentatively determine to have 50% of the β-1,2-INO isomer (a major isomer) to undergo the ring-closure reaction, while the rest of INO does not form epoxides. This leads to a much better agreement with the measured MACR+MVK than the Caltech

mechanism (but still slightly worse than the FZJ simulations). Figure S3-4 shows the model results in comparison to the experiments in Carlsson et al. (2023) and Schwantes et al. (2015). Nevertheless, we regard this as a simplified solution and there are still large uncertainties in this pathway. Future mechanistic studies are needed to better understand the fates of INO. The lower isoprene carbonyl nitrates (ICN) and hydroperoxy nitrates (IPN) in the UCR-ISOP simulations than the other two mechanisms are generally found in the UNC-2010/2012 conditions (OH oxidation with high-$NO_x$, under which $NO_3$ oxidation

of isoprene is also occurring) and the Schwantes-2015 conditions ($NO_3$ oxidation, Fig. S4). This is in large part due to the differed treatment of the $NO_3$ + isoprene pathway in the UCR-ISOP. In particular, the MCM mechanisms assumes that INO exclusively forms ICN, which is also a major source of glyoxal via ozonolysis and OH oxidation (Fig. S2). Thus, the higher simulated ICN in the MCM mechanism is also a major reason for the higher glyoxal predictions. Nevertheless, the UCR-ISOP appears to agree with the Schwantes-2015 experimental data slightly better for ICN and IPN compared to the other two

mechanisms (Fig. S4). Furthermore, the simulated IDHDN and ICHDN in UCR-ISOP are both lower than those predicted by the Caltech mechanism because the INO fates are treated differently. However, it should be noted that these differences stem largely under conditions where later-generation $RO_2$ from $NO_3$ + isoprene react with NO. Such conditions are only prominent in the UNC-2010/2012 experiments. In the real atmosphere, $NO_3$ + isoprene is minimal during daytime when NO may be present. Thus, these differences are less likely to be as significant as found under the laboratory experiments. But as stated

above, this pathway is still highly uncertain and requires future investigation.

Furthermore, we considered that NISOPO2 from isoprene + $NO_3$ undergo rapid self-reactions at a rate coefficient of $5\times10^{-12}$ $cm^3$ molecule$^{-1}$s$^{-1}$, suggested by Schwantes et al. (2015). In a previous study, Kwan et al. (2012) proposed dimer formation from NISOPO2 + NISOPO2 with a branching ratio of 3–4% based on gas-phase measurements. However, Ng et al. (2008) observed a substantial amount of dimers in the SOA from the same experiments, suggesting that the actual dimer formation

branching ratio could be much higher given their very low volatility. In UCR-ISOP, we assume this branching ratio to be 10%,



which leads to a good agreement with the SOA simulation (see next section). This dimer formation pathway from NISOPO2 + NISOPO2 could also partly explain the slightly lower C5-NLV simulation using the UCR-ISOP mechanism under high concentrations (Fig. 3F).

### 3.1.3 SOA Formation Modelling

To evaluate the UCR-ISOP mechanism for SOA formation, we applied the mechanism to model several chamber experiments and compared the simulated SOA with measurements. The chamber experiments include: Kroll-2006 (Kroll et al., 2006), PNNL-2018 (Zawadowicz et al., 2020; Thornton et al., 2020), PNNL-2014 (Liu et al., 2016), Schwantes-2019 (Schwantes et al., 2019), and Ng-2008 (Ng et al., 2008). The detailed model setup can be found in Text S2 in the Supplementary Material. Figure 4 shows the comparisons between modelled isoprene SOA and observations under different conditions. Under low-

$NO_x$ OH oxidation conditions presented in Fig. 4A, a noteworthy consistency between the modelled and measured SOA is evident. Despite a slightly lower bias in the modelled SOA derived from the PNNL-2018 chamber, it lies within a reasonable 50% uncertainty range. This bias can be perceived as reasonable given the underlying uncertainty encapsulating the estimation of C*, measurement and experimental errors, uncertainties in the assumed particle density, and the high sensitivity of SOA mass production to the starting $H_2O_2$/isoprene ratio in these types of experiments (Chen et al., 2023). In comparison to the

simulations from Thornton et al. (2020) for the same experiments, our results show slightly larger discrepancies comparing to the data. This is caused by two main reasons. First, we used the ISOPOOH + OH rate coefficients from the Caltech mechanism which were obtained from carefully performed chamber measurements (Paulot et al., 2009; St. Clair et al., 2016). These rate coefficients are slower than those used by Thornton et al. (2020) by 40% and 6% for the major ISOPOOH isomers. The second reason was described above regarding the different treatments of ISOPOOHOO isomerization rate coefficient and hence the

product distributions. These differences together determine the slightly worse model performance by UCR-ISOP, but as shown in Fig. 4A, the overall model uncertainty within 50% is still reasonable.

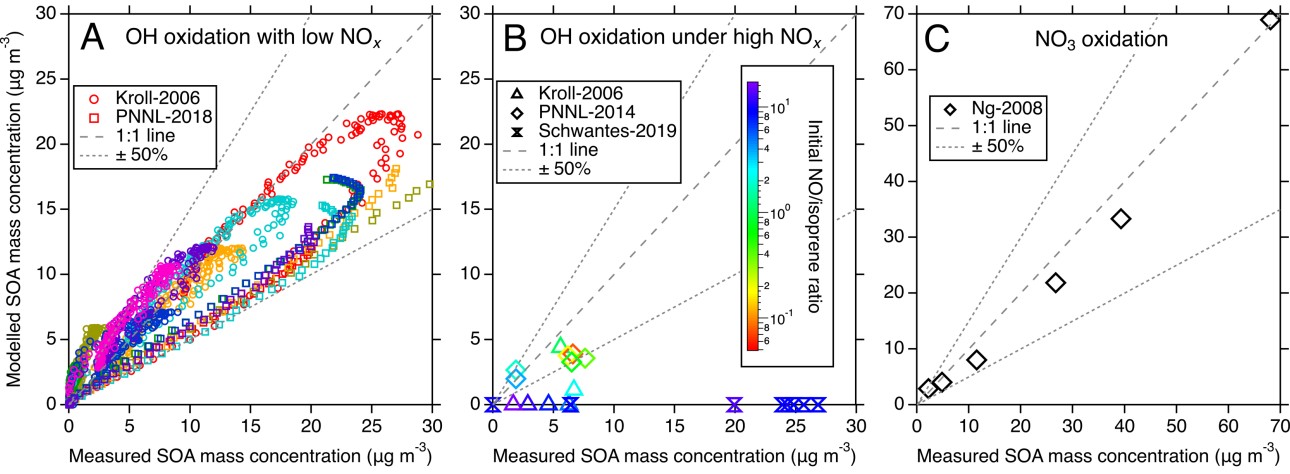



**Figure 4. Comparisons of the modelled and measured SOA in different chamber studies. (A) Modelled vs. measured SOA from the Kroll-2006 (Run 1-9) and PNNL-2018 chamber studies under low-NO$_x$ OH oxidation conditions. Different colors represent different experimental runs. Each marker represents the comparison of real-time SOA mass concentrations. (B) Modelled vs. measured SOA mass concentrations for the Kroll-2006 (Run 9-14), Schwantes-2019 and PNNL-2014 chamber studies under high-NO$_x$ OH oxidation conditions. Each marker represents the maximum SOA mass concentration from a single experiment run and the color scheme represents the initial NO/isoprene concentration ratio. (C) Simulation-data comparisons from the Ng-2008 chamber study under NO$_3$ oxidation conditions. Each marker represents the maximum SOA mass concentration from a single experiment run. In all panels, the dashed line indicates a 1:1 correspondence, and the dotted line delineates a 50% uncertainty boundary.**

Contrastingly, the high-NO$_x$ OH oxidation conditions outlined in Fig. 4B present a complicated case in the SOA simulations. When the initial NO/isoprene concentration ratio is relatively low (<2), which corresponds to PNNL-2014 and part of the Kroll-2006 experiments, the model predicted SOA is generally consistent with the measurements within 50%, similar to the results under low-NO$_x$ conditions shown in Figure 4A. Under these conditions, the predicted SOA composition is a mixture of both the low-NO$_x$ products like IDHDP and ICPDH, and high-NO$_x$ products like IDHPN, IDHDN. Lower initial NO/isoprene ratio enhances the contribution from the low-NO$_x$ products, leading to the overall reasonable simulation-measurement agreement. However, when the initial NO/isoprene ratio is relatively high (> 2), which corresponds to Schwantes-2019 and some of the Kroll-2006 experiments, there exists a discernible underestimation in the modelled SOA formation, with the simulations showing in principle negligible SOA formation. To rule out that this may be due to the uncertainties in volatility calculations, we estimate the upper limit of the SOA mass concentrations by assuming that all the included low-volatility products (e.g., IDHDN, IDHPN, ICHNP, IDHCN and ICHDN) can entirely partition to the particle phase (Fig. S5). But this still significantly underestimates the SOA mass concentrations. This stark deviation from measurements is not only for the UCR-ISOP mechanism but is a common theme observed in other compared chemical mechanisms, as shown in Fig. S5. This pronounced discrepancy unveils a substantial gap in our current understanding of isoprene oxidation under the high-NO$_x$ conditions, necessitating a concerted focus on future studies to unravel the complexities therein.

For the NO$_3$ oxidation of isoprene, as the simulations shown in Fig. 4C, there is reasonable consistency between the modelled and the measured SOA mass concentration, with dimer, IDHDN, and IHPDN as the primary contributors (Fig. S6). For the Caltech and MCM-UW mechanisms (Thornton et al., 2020), the dominant contributors are IDHPN, IDHDN and ICHNP. The difference in SOA species largely comes from the fates of NISOPO2. In the Caltech mechanism, the dominant sink of NISOPO2 is to react with NO$_3$ and itself to form INO, IHN, and ICN, which will later be oxidized into IDHPN, IDHDN or ICHNP. A small fraction of NISOP2 is consumed by HO$_2$ to form hydroperoxide nitrates and other species like MVK and MACR. In the UCR-ISOP mechanism, the formation of dimer from NISOPO2 + NISOPO2 as described above, also greatly contribute to the SOA under this experimental condition. In a recent study, Graham et al. (2023) showed that SOA from isoprene + NO$_3$ exhibit lower volatility than that from α-pinene + NO$_3$, supporting that dimers are largely present in the isoprene + NO$_3$ SOA.

In addition to the gas-phase formation mechanisms for the low-volatility products, the other potential major uncertainty in simulating isoprene SOA from these chamber experiments lies in the C* calculations. As described in ***section 2.3***, The C*



values of the low-volatility species are calculated from the vapor pressure, which can be estimated using EVAPORATION
(Compernolle and Ceulemans and Müller, 2011) and SIMPOL.1 (Pankow and Asher, 2008). Both EVAPORATION and
SIMPOL.1 are group contribution structure-activity relationships, but EVAPORATION includes proximity-based functional
group interactions, so responds to differences in the locations of functional groups, while SIMPOL.1 does not vary based on
functional group locations. In the results shown in Fig. 4, we used EVAPORATION to estimate the vapor pressures, using the
lowest vapor pressure for all the possible isomers of each low-volatility species. This introduces some uncertainty, as the
lowest-volatility isomers are not necessarily the most dominant isomers, which are not always obvious due to mechanism
reduction. To investigate how the selections of isomeric structures and vapor pressure estimation methods could affect the
simulated SOA, we compare the simulated maximum SOA mass concentrations for the experiments shown in Fig. 4
(experiments with NO/isoprene > 2 excluded for this comparison) using different vapor pressure estimation methods and (in
the case of EVAPORATION) isomers with the higher-bound $C^*$ vs. those with the lower-bound $C^*$. The comparison results
shown in Fig. S7 suggest that the model predicted SOA is generally lower than the measured values, especially when the
higher-bound $C^*$ are adopted. When the lower-bound $C^*$ are used (as used in the simulations shown in Fig. 4), the model
prediction is within 50% comparing to the measurements. Using higher-bound vapor pressures in EVAPORATION or using
SIMPOL.1, simulated SOA is lowered by ~ 20%. This highlights the needs to better estimate the vapor pressures of
multifunctional oxidation products in SOA as they may lead to as great uncertainties as those from the less constrained
chemical mechanisms.

## 3.2 Simulation of Field Observations Using the Multiphase Isoprene Chemical Mechanism

### 3.2.1 Model Setup

To further evaluate the multiphase isoprene mechanism and understand the impacts of the various pathways on isoprene SOA
formation under atmospheric conditions, we performed 0-D kinetic box model simulations for the 2013 SOAS campaign and
compared our model results with the field observations. The field site information can be found in previous literature (Xu et
al., 2015; H. Zhang et al., 2018; Lee et al., 2016; Nguyen et al., 2015a). In the F0AM-WAM model setup, meteorological
parameters such as temperature, pressure, RH, and boundary layer height were directly obtained from the measurements.
Photolysis rates in the model were calculated from real-time solar zenith angle (not adjusted by cloud coverage). Model inputs
including the gas-phase concentrations of isoprene, OH, $HO_2$, NO, $NO_2$, $NO_3$, and $O_3$, as well as the mass concentrations of
total submicron organic aerosols, LWC, and inorganic ions were all constrained by measurements that were averaged hourly
throughout the campaign. In dealing with missing data, for instances where data was missing for less than 6 hours, linear
interpolation was applied. In cases where the missing data spanned longer, we used an average diurnal profile, derived from
measurements taken throughout the entire field campaign. The submicron organic aerosol mass concentrations measured by
time-of-flight aerosol mass spectrometry (AMS) were used to calculate gas-particle partitioning based on organic absorptive
equilibrium (Pankow, 1994); the inorganic ion concentrations also from AMS measurements were used to estimate aerosol



acidity (Song et al., 2018); the LWC data were used to calculate aqueous uptake of water-soluble compounds (Wania et al., 2015). The AMS-derived positive matrix factorization (PMF) for IEPOX-SOA was used to compare with our simulated isoprene SOA (Hu et al., 2015). Furthermore, molecular-level measurements of gas-phase isoprene products measured in real-time by time-of-flight chemical ionization mass spectrometry with $CF_3O^-$ ionization ($CF_3O^-$-CIMS) (Nguyen et al., 2015a),

and particle-phase isoprene oxidation products measured by offline thermal desorption two-dimensional gas chromatography time-of-flight mass spectrometry (TD-GC×GC-MS), in-situ semi-volatile thermal desorption aerosol gas chromatography mass spectrometry (TAG-MS), and the iodide-adduct CIMS with a filter inlet for gases and aerosols (FIGAERO) were also used to compare with the model simulations (Lee et al., 2016; H. Zhang et al., 2018; Isaacman-Vanwertz et al., 2016).

Both dry and wet depositions are parameterized and incorporated in the box model. For each species, the dry deposition

velocity (cm s$^{-1}$) is assumed to follow a diurnal pattern proportional to the cosine of the solar zenith angle with peak value estimated using the parameterization method illustrated in Nguyen et al. (2015a). The dry deposition velocity for particles is assumed to be 0.2 cm s$^{-1}$ (Farmer and Boedicker and Debolt, 2021). The dry deposition rate (s$^{-1}$) for each species is the ratio of its dry deposition velocity to boundary layer height. For the wet deposition, Bi and Isaacman-Vanwertz (2022) illustrated that the wet deposition lifetime for one species can be simply estimated only from its Henry's law constant, $H_{aq}$. The detailed

precipitation information like droplet distribution and precipitation intensity has little influence on the wet deposition lifetime. Thus, in our model, we calculated the wet deposition lifetime for all species based on $H_{aq}$ values and apply the corresponding wet deposition rate (i.e., first order loss at a rate of 5.5×10$^{-5}$ s$^{-1}$ for the most soluble gases) when precipitation is observed. The $H_{aq}$ values were estimated from EPI Suite (Card et al., 2017). The estimated $H_{aq}$ were also used for modelling aqueous-phase uptake. Given the difficulty in quantifying the influence of advection on local concentrations in the 0-D model, a first-order

dilution rate was added to all species to account for potential mixing and ventilation. The diurnal variation of the dilution rate was scaled based on boundary layer height (Kaiser et al., 2016) with the time-dependent scaling factors determined such that the modelled MACR + MVK concentrations could approximately agree with the measurements. This is based on the assumption that the MACR and MVK can be reasonably simulated by the Caltech isoprene mechanism (Zhang et al., 2022). However, as discussed above, this treatment could have larger uncertainties for nighttime dilution rate estimation owing to the

potential overprediction of MACR + MVK from isoprene NO$_3$ oxidation by UCR-ISOP. Nevertheless, the overprediction is only within a factor of two (Fig. S3) and nighttime isoprene concentration is very low at SOAS. Thus, this is likely less critical than the other uncertainties discussed in this work.

During the SOAS campaign, because RH is usually in the moderate to high level (50–100%) and many oxidation products from isoprene are relatively water-soluble (Fig. S8), the aqueous uptake of soluble compounds were considered for the species

with $H_{aq}$ larger than 1×10$^7$ M atm$^{-1}$, including all the above-mentioned low-volatility species as well as other smaller water-soluble products, such as glyoxal. Furthermore, because the particle phase state is very important for the reactive uptake of IEPOX (Y. Zhang et al., 2018), the average O:C ratio (derived from the AMS measurements), organic mass to sulfate ratio and ambient RH were used to determine the occurrence of phase separation behavior (Schmedding et al., 2020). Aerosols were



found to be internally mixed for most of time during the SOAS campaign at ground level (Fig. S9) and this assumption was
adopted in the calculation of reactive uptake coefficients.

### 3.2.2 SOAS Simulation Results

With the hourly constraints of meteorological conditions, isoprene, and major oxidants (Fig. S10), the model is well suited to
simulate isoprene chemistry at the SOAS ground site. Because a portion of the $HO_2$ concentrations were from prior CMAQ
simulations rather than measurements, extra verification was conducted by comparing the modelled gas-phase $H_2O_2$ (mostly
from $HO_2 + HO_2$) with the measurements (Fig. S11), which shows reasonable agreement. This suggests that the initial isoprene
oxidation chemistry and ISOPOHOO's various unimolecular and bimolecular fates (e.g., reacting with NO and $HO_2$) are
expected to be well represented. This setup should also lead to reasonable simulations of the first- and second-generation major
products whose chemistry has been well studied. However, when comparing these gas-phase products between the simulations
and quantitative measurements, especially for ISOPOOH+IEPOX and IHN, the model overpredict their concentrations by
factors of 1.8 and 1.9 on average and reaching 2.7 and 3.3 at daily peak, respectively (Fig. S12 and S13). This overprediction
is not only produced by UCR-ISOP, but also by the Caltech mechanism because these two mechanisms predict almost identical
gas-phase concentrations. It should be pointed out that the loss of these species via reactive uptake onto acidic/aqueous particles
(for IEPOX and IHN) has already been considered in UCR-ISOP using kinetic information from the literature (Vasquez et al.,
2020; Pye et al., 2013). For IEPOX reactive uptake, we tested different $H_{aq}$ values that have been reported in prior studies
(ranging from $1.9 \times 10^7$–$4 \times 10^8$ M atm$^{-1}$), but the uptaken amount of IEPOX is not very sensitive to this value and this uncertainty
does not help resolve the differences. Thus, the discrepancy must suggest additional loss pathways for these species, such as
loss via cloud interactions, which is not considered in the current model. Alternatively, better quantification of these key gas-
phase intermediates is needed.

The simulated isoprene SOA diurnal medians from three general categories (i.e., IEPOX-SOA, C5-LV, and C5-NLV) are
shown in Fig. 5A–C along with the respective molecular-level measurements during the 2013 SOAS campaign. The C5-LV
and C5-NLV represent C5 low-volatility species without and with nitrogen, respectively. The detailed time series comparison
throughout the field campaign can be seen in Fig. S14. As shown in Fig. 5A, the measurements from FIGAERO-CIMS (the
"2-MT data") are the summation of the estimated concentrations for chemical formulas of $C_5H_{12}O_4$, assumed to represent 2-
MT and $C_5H_{10}O_3$, assumed to represent C5-alkenetriols; those from the filter-based TD-GC×GC-MS are the non-OS IEPOX-
SOA data, including 2-MT, C5-alkenetriols, and other species that well correlate with them (H. Zhang et al., 2018). However,
it has been suggested that the IEPOX-OS may partly decompose to 2-MT and C5-alkenetriols during the thermal desorption
process in GC (Rattanavaraha et al., 2016; Cui et al., 2018). Thus, the TD-GC×GC-MS measurements could represent the total
IEPOX-SOA to some extent. The diurnal variations from both measurements are characterized by a peak in the afternoon and
a nadir in the morning. Our modelled IEPOX-SOA nicely replicate this diurnal trend and the magnitude, demonstrating its
proficient capacity in mirroring the IEPOX reactive uptake pathway. In contrast to the comparison shown in Fig. 5A, the TAG-





MS measurements exhibit a much higher concentrations especially during nighttime and morning (Fig. S15), despite that the TAG-MS analysis principle is essentially the same as that by the TD-GC×GC-MS. The discrepancies in these measurements suggest that the 2-MT and C5-alkenetriols tracers are likely partly decomposed from other tracers (e.g., IEPOX-OS) to different extents, highlighting the needs to better quantify these important IEPOX-SOA tracers in future work.

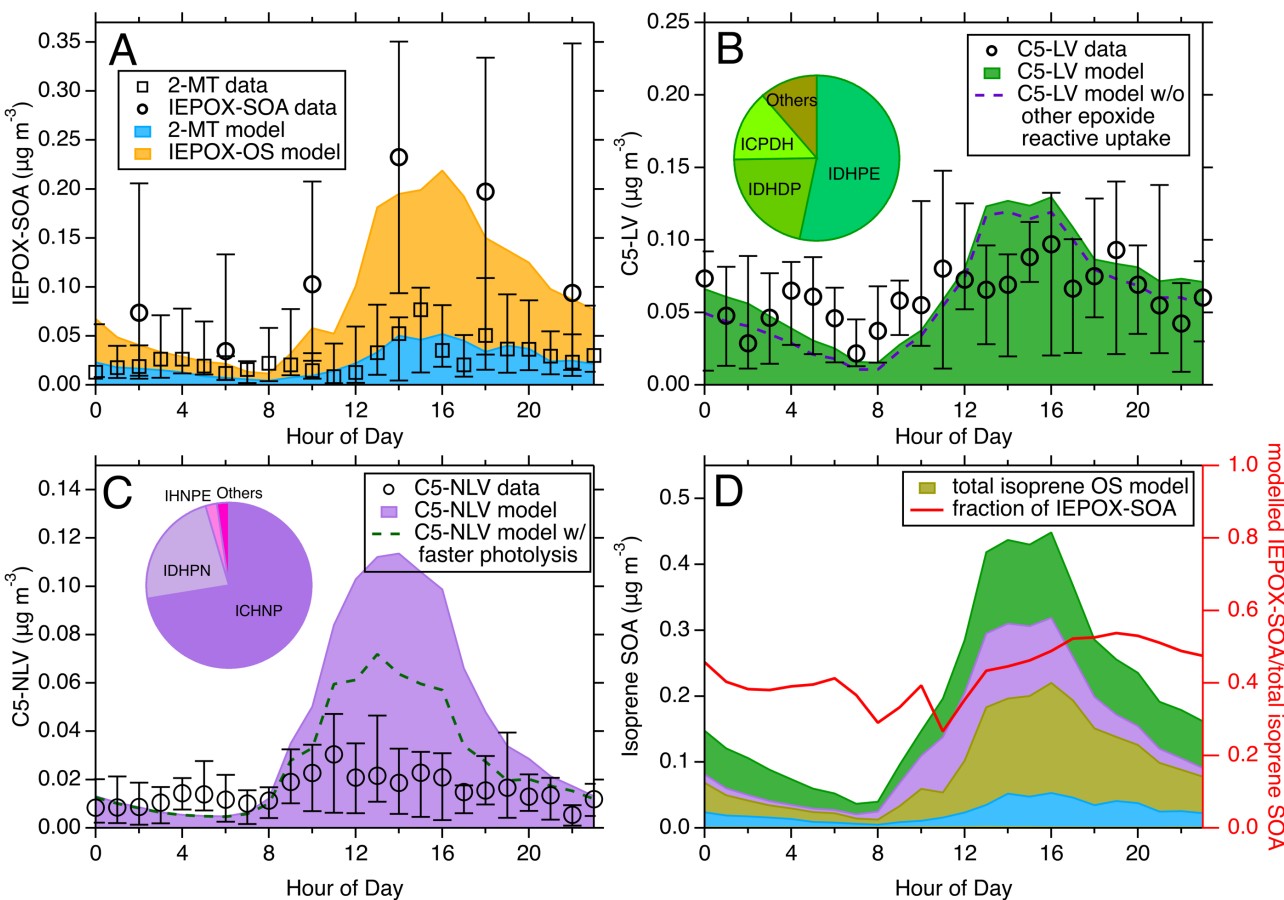


**Figure 5.** **The diurnal trend of the modelled and measured isoprene SOA from different pathways during the 2013 SOAS campaign. (A) The modelled diurnal trends of IEPOX-SOA from the reactive uptake pathway. The measured filter-based isoprene SOA by TD-GC×GC-MS and the measured 2-MT by FIGAERO-CIMS are presented. (B) Comparison between the modelled and measured total C5-LV, the latter of which is derived from a detailed summation of specific molecular species measured by FIGAERO-CIMS.**
**The dashed line corresponds to the model results when the reactive uptake of other epoxides is not considered. (C) Comparison between the modelled and measured total C5-NLV. The dashed line corresponds to the model results when rapid photolysis of C5-NLV species containing a –OOH group is assumed. The inserted pie charts in (B) and (C) show the relative contributions of several major species to the respective SOA categories. (D) The total modelled isoprene SOA from different pathways (left $y$-axis). The "ISOP-OS" includes both the IEPOX- and other epoxides-derived OS; the colors blue, purple, and green represent the same**
**categories as shown in (A)–(C). The diurnal fraction of IEPOX-SOA in total modelled isoprene SOA is shown with the right $y$-axis. For all the measurement data, the whiskers indicate the 25th and 75th percentiles, respectively.**





Figure 5B and 5C show the median diurnal variations of modelled C5-LV and C5-NLV, respectively. The measured C5-LV and C5-NLV are determined from the sum of many molecular species measured by the FIGAERO-CIMS with chemical
formulas of $C_5H_{8-12}O_{5-7}$ (C5-LV) and $C_5H_{7-11}O_{5-9}N$ (C5-NLV) (Lee et al., 2016; H. Zhang et al., 2018). It should be noted that some of these formulas may not be all from isoprene oxidation. For instance, $C_5H_8O_5$ could also be 3-hydroxyglutaric acid from monoterpene oxidation (Claeys et al., 2007). In addition, the quantification of these species in the FIGAERO-CIMS data could have high uncertainties. For example, the C5-NLV species were quantified using an IHN isomer with the highest sensitivity as the surrogate standards (Lee et al., 2016). Thus, the quantified C5-NLV is considered the lower limit. For C5-
NLV species which have lower sensitivity such as carbonyl nitrates, this quantification approach will underestimate the concentrations. These above-mentioned issues could lead to uncertainties in quantifying the C5-LV and C5-NLV mass concentrations. Nevertheless, these measurements are likely still the best quantitative data available from field measurements. The simulated C5-LV shown in Fig. 5B exhibit similar diurnal trends and magnitude as the measurements, while the C5-NLV shows a more significant discrepancy between the modelled and the measured mass concentration (Fig. 5C), in part due to the
quantification uncertainties mentioned above. It is interesting to note that in the C5-LV simulations, ICPDH, ITHC and IDHPN in these species can partly be contributed by the reactive uptake of epoxides other than IEPOX. As shown in the gas-phase simulations (Fig. S16), although the other epoxides are smaller than IEPOX, they may also be important with the largest epoxides being ICPE from ISOPOHOO isomerization and ICHE from HPALD oxidation. This underscores the need to further study the multiphase fates of these previously less-studied epoxides. In addition, the C5-NLV simulations may also be greatly
affected if hydrolysis and photolysis rates are treated differently. For example, as shown in Fig. 5C, if C5-NLV species containing –OOH groups are allowed to undergo the rapid photolysis those formed in the low-$NO_x$ conditions are, the simulated C5-NLV may be largely reduced. To gain further insights into the roles of the low-volatility pathways in the isoprene SOA formation, the modelled molecular contributions for each SOA category throughout the SOAS field campaign are investigated. In the C5-LV category, IDHPE and IDHDP are the two largest contributors (see the pie chart insert in Fig. 5B), both originating
from the oxidation of ISOPOOH under low-$NO_x$ conditions. Interestingly, despite that IDHPE and IDHDP are predicted to have large concentrations in SOAS isoprene SOA, their chemical formula ($C_5H_{12}O_6$, assuming IDHPE opens the epoxide ring in the particle phase to form the hydroperoxyltetrols) was found to be very low in the FIGAERO-CIMS measurements (D'ambro et al., 2017). It should be noted that rapid photolysis of these hydroperoxide compounds have already been considered in the model. Thus, we suspect that additional multiphase or bulk-phase reactions also readily take place that further
transform these labile species into more stable oxygenated compounds (e.g., products reported by Jaoui et al. (2019)). ICPDH ($C_5H_{10}O_5$) is the third largest C5-LV species that is formed from IEPOX + OH oxidation followed by bimolecular reaction with $HO_2$. In the C5-NLV category, ICHNP ($C_5H_9NO_7$) primarily from IHN + OH and subsequent $RO_2$ (ISOPNOO) isomerization emerges as the largest contributor (> 70%, see the pie chart insert in Fig. 5C). This chemical formula has been shown as a major particle-phase organic nitrates with timeseries consistent with isoprene SOA during SOAS (Lee et al., 2016).
The other main C5-NLV species, IDHPN ($C_5H_{11}NO_7$) can be formed from both ISOPNOO + $HO_2$ and ISOPOOHOO + NO (see Fig. 1).




In Fig. 5D, we present the diurnal variations of the total isoprene SOA derived from both the low-volatility and reactive uptake pathways, offering a holistic perspective on the isoprene SOA composition and concentrations stemming from different formation mechanisms. This comparison suggests that the explainable non-IEPOX fraction accounts for ~57% of total

simulated isoprene SOA during SOAS throughout the day, which is unexpected and highlights the importance to better understand the reaction pathways in more detail. Notably, this fraction is approximately consistent with a previous laboratory study (Liu et al., 2015), but the results shown in the present work are for realistic atmospheric conditions. In addition, we also compare the total simulated low-volatility SOA (C5-LV + C5-NLV) in correlation with the IEPOX-SOA with hourly resolution (Fig. S17). It turns out that the isoprene SOA from these two formation pathways correlated reasonably well ($R^2 = 0.80$).

Interpreting from this strong timeseries correlation and based on the way PMF works in deconvoluting organic aerosol sources (Zhang et al., 2007; Lanz et al., 2007; Ulbrich et al., 2009), we suggest that the AMS-PMF analysis may not always effectively separate the IEPOX-SOA from the other isoprene SOA, despite that prior studies have reported a specific AMS-PMF factor for the ISOPOOH-derived SOA which is quite different than the well-known IEPOX-SOA factor (Riva et al., 2016). Therefore, the IEPOX-SOA factor from AMS-PMF, which was previously considered to represent SOA only from the IEPOX reactive

uptake pathway, could partly include isoprene SOA from the LV pathways.

However, a discernible underestimation of our modelled total isoprene SOA is present, when compared to the IEPOX-SOA factor (Fig. S15), suggests additional SOA formation pathways from isoprene oxidation in the atmosphere that our mechanism does not include. In chamber experiment simulations discussed above, we failed to simulate SOA formation under initial NO/isoprene ratio higher than 2. But during the SOAS campaign, the Centreville site is always under low-$NO_x$ conditions with

this ratio usually lower than ~0.1 (Fig. S10). Thus, it is unlikely that this unrepresented SOA formation explains the model underestimation. In our model, we extensively included pathways that lead to multifunctional low-volatility products which retain the isoprene C5 backbone. Thus, it is possible that some fragmentation products (< C5) may also contain multiple functional groups through further oxidation and contribute to SOA formation. For these species, we only included SOA from HMML/MAE as they are well-studied isoprene SOA precursors (Lin et al., 2013; Nguyen et al., 2015b). This could partly

explain the observed model-observation discrepancy. Moreover, although dimer formation from $RO_2 + RO_2$ reactions are considered in the model in both the daytime and nighttime pathways with rapid reaction rate coefficients and high branching ratios, they are predicted to be low under the SOAS conditions. However, prior studies suggested that the formation of dimers from isoprene $RO_2$ and monoterpene $RO_2$ may be prominent under conditions like the SOAS site (Tiszenkel and Lee, 2023). This process is not included in our mechanism because the monoterpene chemistry is not explicitly described. To better

understand the isoprene SOA molecular composition from these pathways, especially in atmospheric aerosols, future research is certainly warranted.

It is also remarkable to note that the predicted C5-NLV mass concentration is nearly as high as (ratio ~0.91) that of that from the C5-LV category at daily maximum during SOAS, despite that the field site is an isoprene-dominant forest area with low-$NO_x$. For reference, the major gas-phase intermediates for C5-NLV, IHN, are about a factor of 10 lower than those for C5-

NLV, ISOPOOH + IEPOX (Fig. 6A–B). The strong contrast suggests that the IHN oxidation pathway is much more efficient



in producing LV SOA than the ISOPOOH/IEPOX + OH pathway. We suggest that this is resulted from several effects. First, the oxidation of ISOPOOH and IEPOX produce C5-LV $RO_2$ (i.e., ISOPOOHOO and IEPOXOO) at smaller branching ratios than C5-NLV $RO_2$ (ISOPNOO) production from IHN + OH (Wennberg et al., 2018). This leads to a reduced difference between the production rates of the C5-LV $RO_2$ and the C5-NLV $RO_2$ to a factor of ~4 (Fig. 6C–D). Furthermore, ISOPNOO

is more effective to be converted to ICHNP (dominant C5-NLV species) under the SOAS conditions, in comparison with ISOPOOHOO and IEPOXOO to IDHPE, IDHDP, and ICPDH. As shown in Fig. S18, the pseudo first-order rates of $RO_2$ against bimolecular reactions ($k_{RO2,1st}$) during SOAS estimated using $HO_2$ and NO measurements is < 0.02 s$^{-1}$ during most time of day. Thus, the unimolecular isomerization of ISOPNOO (~ 0.04–0.08 s$^{-1}$) outcompetes its bimolecular reactions and directly produces the most abundant C5-NLV species, ICHNP at 100% yield. In contrast, the unimolecular isomerization rate constant

of ISOPOOHOO is similar to $k_{RO2,1st}$, while that of IEPOXOO is fast but produce C5-LV products at much lower yields (Wennberg et al., 2018). As a result, the produced C5-NLV/C5-LV ratio in the gas phase is further reduced to a factor of only ~2 (Fig. 6E–F). Lastly, the presence of LWC brings particle-phase C5-NLV and C5-LV concentrations even closer, because LWC more prominently enhances SOA formation for the C5-NLV species. As shown in Fig. S8, the C5-NLV species have the highest $H_{aq}$ values. This leads to the high sensitivity of this category to LWC. In contrast, the LWC allowing for IEPOX

reactive uptake diminishes formation of some of C5-LV species from IEPOX oxidation. As a result, in sensitivity analysis shown in Fig. 6G–H, one can see that the C5-NLV mass concentration and ratio over C5-LV both substantially increase as LWC is enhanced from 0 to the SOAS ambient level. These comparisons indicate that formation of organic nitrates in SOA can be important even for low-NO$_x$ environments.





**Figure 6.** The comparisons of the C5-LV and C5-NLV formation pathways. (A)–(B): The modelled diurnal concentrations of ISOPOOH + IEPOX (precursors for C5-LV) and IHN (precursors for C5-NLV). (C)–(D): The modelled diurnal RO₂ production rates for C5-LV RO₂ (i.e., ISOPOOHOO and IEPOXOO) and C5-NLV RO₂ (i.e., ISOPNOO). (E)–(F): The modelled diurnal concentrations of gas-phase C5-LV and C5-NLV. In (A)–(F), the simulations are for gas phase only. (G)–(H): influence of LWC on particle-phase C5-LV and C5-NLV mass concentrations. Simulated scenarios include LWC = 0 µg m⁻³ (red), LWC set to be a factor of 10 lower than actual concentrations (olive), and LWC from actual concentrations (blue). In (H), "R" represents the ratio of C5-LV and C5-NLV at daily maximum (averages from 12:00 to 16:00, local time). Throughout the figure from top to bottom, the ratios of the corresponding N-containing over non-N-containing products increase from 0.1 to 0.91 under SOAS conditions.



## 4. Future Directions and Atmospheric Implications


While the molecular-level understanding of isoprene oxidation chemistry has improved significantly, it is still challenging to include all the process into a multiphase chemical mechanism for laboratory and atmospheric SOA predictions. This work first presents such a condensed chemical mechanism for modelling gas-phase isoprene oxidation chemistry as well as SOA with the major molecular products represented. Our condensed mechanism provides a substantial step toward the improved model

estimation of isoprene-derived SOA, including both the multigenerational oxidation leading to low-volatility products and the reactive uptake pathways. In the process of developing and evaluating the new mechanism by comparing with other mechanisms and data from chamber studies and field measurements, it is recognized that there remain significant uncertainties in understanding isoprene oxidation chemistry and SOA formation. Thus, we consider this mechanism to be a starting point with flexibilities for future updates.

Among the many uncertainties, a few major ones are summarized here. First, there are large discrepancies in isomerization rate coefficients for some key isoprene $RO_2$ and RO species between different measurements and between experiments and computational calculations. These rate coefficients may be crucial for determining the $RO_2$ and RO fates and hence product distributions, especially under atmospheric conditions where unimolecular isomerization (for $RO_2$) can be more important. An example we show in this work is that the difference in $RO_2$ fates plays a significant role controlling the C5-NLV composition

and formation efficiency: In the SOAS simulations, the ISOPNOO unimolecular isomerization outcompetes its bimolecular reactions to directly produce the most abundant C5-NLV species, ICHNP at 100% yield. Instead, under the laboratory experimental conditions, bimolecular reactions of ISOPNOO likely dominate its fate and produce other C5-NLV species at smaller yields. These striking differences highlight the challenge to mimic atmospheric oxidation conditions in laboratory experiments and the distinct $RO_2$ fates can significantly shift the product distributions. Besides, gas-phase dimer formation

from $RO_2 + RO_2$ reactions should be better understood. This is not a need only for the isoprene chemistry, but for other VOC systems as well (e.g., monoterpenes). It can be even more complex but important when $RO_2 + RO_2$ reactions occur across different VOC systems. Moreover, we suggest that the gas-phase mechanisms for isoprene + $NO_3$ as well as the high-$NO_x$ pathways are not well understood, in terms of how and what low-volatility products are formed. In addition, we show that our model does not accurately predict the atmospheric concentrations of major gas-phase products such as ISOPOOH, IEPOX,

and IHN in the SOAS field campaign. We regard this to be a lack of understanding of their missing sinks, rather than their formation chemistry, because the mechanisms and kinetics for their production are likely well understood from prior laboratory studies. Investigating the missing sinks could be crucial for improving our understandings of areas such as cloud processes.

Regarding isoprene SOA formation, a major uncertainty lies in the gap between the predicted total SOA and the measurements from a variety of techniques (e.g., AMS, FIGAERO-CIMS, and GC-based techniques). It is certainly crucial to resolve the

differences between these measurements and examine the possible decomposition processes during analyses. We also suggested that part of the gap is due to the SOA species with smaller than C5 not represented in the current mechanism. But it should be noted that the gap is more significant during nighttime and morning, especially when comparing the simulations

with measurements by TAG-MS and AMS, which exhibit a smaller diurnal pattern than the model (e.g., comparing Fig. S15 and Fig. 5D). The source of this nighttime SOA is not well described by the model, warranting future investigation. In addition,

the vapor pressure estimation of the low-volatility species could also introduce uncertainty. This uncertainty may be a greater challenge in cases where isomers are lumped in a condensed mechanism. Furthermore, prior studies have extensively studied the reactive uptake of IEPOX (Surratt et al., 2010; Lin et al., 2012; Lin et al., 2013; Nguyen et al., 2014; Y. Zhang et al., 2018). Here, we show that if the other epoxides formed from isoprene oxidation undergo similar reactive uptake reactions, they may also contribute to SOA formation. These epoxides should be more thoroughly studied in future research. Lastly, significant

uncertainties remain for the particle-phase reactions. In the current mechanism, we simplified the photolysis for hydroperoxides and hydrolysis for organic nitrates by using the same photolysis and hydrolysis rate coefficients, respectively. But we also show that the total isoprene SOA mass concentrations and compositions could be greatly affected by these parameters. In addition, other particle-phase reactions such as oxidative aging and accretion are not well constrained but are important to bridge molecular-level measurements and model predictions.

Despite the uncertainties, the model can reasonably predict the mass concentration and composition of isoprene SOA in the SOAS field campaign, estimating the contributions from different pathways in the ambient environments. Our model results also highlight that the low-volatility pathways contribute greatly to the isoprene SOA formation. We expect that their importance could be even higher in the future given the emissions of sulfur and nitrogen are reduced. Implementing the multiphase mechanism in air quality models in future studies may provide new insights into isoprene SOA chemistry in the

regional and global scales.

**Appendices**

Detailed supplementary results are provided in the Supplementary Material.

**Code Availability**

The condensed mechanism (UCR-ISOP) can be provided upon request to the corresponding author.

**Author Contribution**

CS and HZ designed and performed the simulations. JT, JS provided insights into the laboratory experiments and chemical mechanisms. The F0AM-WAM model was originally developed by JT's research group. XY prepared part of the mechanism development. CB and GIV discussed about the results and contributed part of the model simulation. CS and HZ prepared the

manuscript with contributions from all co-authors.



**Competing Interests**

The authors declare that they have no conflict of interest.

**Acknowledgements**

This work was supported by the U.S. National Science Foundation (AGS-2037698). PNNL experiments and author were
supported by the Atmospheric System Research (ASR) program as part of the DOE Office of Biological and Environmental
Research under PNNL project 57131. Pacific Northwest National Laboratory is operated by DOE by the Battelle Memorial
Institute under Contract DE-A06-76RLO1830.



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
