# Peer review of "Observation-Constrained Kinetic Modelling of Isoprene SOA Formation in the Atmosphere"

_EGUsphere, 2024_

## Author Comment (AC1)

Dear Editor and the reviewers:

We appreciate your comments and suggestions on our initial submission and the opportunity to revise the manuscript. Based on your comments and requests, we have amended the relevant parts in the manuscript. Our point-to-point responses to the reviewers' comments are itemized below. To guide the review process, we have copied the reviewers' original comments in italic black text. Our responses are in regular-font red text, and the modified text in the revised manuscripts is in regular-font blue text.

Please feel free to let us know if you have further questions or comments.

Sincerely,

Chuanyang Shen and Haofei Zhang

*Reviewer #1:*

*This manuscript describes the evaluation of an isoprene oxidation mechanism designed to be compact enough for large-scale (regional to global) modeling while representing the pathways of secondary organic aerosol (SOA) formation in sufficient detail to simulate ambient and chamber observations across a wide range of conditions. The manuscript then compares box-modeled SOA using the new mechanism to a series of chamber experiments in both high and low NOx conditions as well as ambient observations from the SOAS campaign. Crucially, the mechanism enables comparisons of specific SOA constituents -- e.g. IEPOX-SOA, low volatility nitrogen-containing compounds, etc. -- with such speciated observations, allowing the authors to show that some subcategories compare well while others don't; notably, C5 nitrogen-containing low-volatility compounds seem overestimated compared to ambient observations, but SOA in high-NOx chamber experiments was underestimated, which poses an intriguing conundrum. Also of note, the commonly used AMS-PMF factor for IEPOX-SOA seems likely to also incorporate other non-IEPOX isoprene-derived SOA.*

Response:

Thank you for your insightful comments and the opportunity to clarify the experimental conditions underpinning our observations. Specifically, the ambient SOAS observations referenced in our study represent low-NOx conditions, characterized by a NO to isoprene ratio lower than ~0.1 (Fig.S10). This is a key distinction because these conditions fundamentally differ from those in the high-NOx chamber experiments with an initial NO/isoprene ratio higher than 2. Therefore, the comparison of ambient observations with high-NOx chamber experiments involves inherently different chemical environments and reaction pathways. The conundrum pointed by the reviewer is due to the different measurements: the underestimation of high-NOx chamber SOA is by comparing total SOA mass concentration between our model and measurements; the overestimation of C5-NLV at SOAS is by comparing the speciated compounds, the latter of which could arise from the measurement uncertainties and the model treatment of particle chemistry, which is shown in the manuscript as the following:

L556-559: "… the C5-NLV species were quantified using an IHN isomer with the highest sensitivity as the surrogate standards (Lee et al., 2016). Thus, the quantified C5-NLV is considered the lower limit. For C5-NLV species which have lower sensitivity such as carbonyl nitrates, this quantification approach will underestimate the concentrations."

L567-570: "In addition, the C5-NLV simulations may also be greatly affected if hydrolysis and photolysis rates are treated differently. For example, as shown in Fig. 5C, if C5-NLV species containing –OOH groups are allowed to undergo the rapid photolysis those formed in the low-NOx conditions are, the simulated C5-NLV may be largely reduced."

*Overall, I found I had a lot of questions about the specifics of the mechanism which couldn't be answered with the level of detail provided. The authors should make the full mechanism available*

*in the supplement or via a readily accessible public link so as to allow readers to answer these questions without having to request (and attempt to interpret) mechanism files. (Providing the tables of species and their SMILES strings in the SI, though, is a big plus!). The discussions of uncertainties, especially in the NO3 pathway (L 322-350) and in C\* (L 412-430), are particular highlights -- I was very glad to see these included and described as well as they were. The manuscript is well-written and will be a valuable addition to the literature, and should be published after addressing some of the following concerns and providing the mechanism itself.*

Response:
We sincerely thank you for your constructive feedback and for recognizing the strengths of our work. Regarding the full mechanism and its accessibility, as our paper is under the peer review process and updates may be required depending on the review outcome, we have not publicly released the full mechanism. However, we fully intended to provide a public link to make the complete mechanism available once the paper is accepted for publication. This will enable interested readers to explore the mechanism in detail and use it freely without the need to request additional information. In the revised manuscript, the code availability with the link has been added:

L708-709: "The condensed mechanism (ISOP-UCR) and associated codes in the F0AM format can be found at https://github.com/zhangucr/UCR-ISOP.git."

*First, it's misleading to suggest that this UCR-ISOP mechanism represents a unique development in the incorporation of detailed isoprene chemistry into global models, or that previous global modeling studies have neglected to include detailed isoprene chemistry leading to SOA formation. Stadtler et al. (DOI 10.5194/gmd-11-3235-2018), Bates & Jacob (DOI 10.5194/acp-19-9613-2019), and Müller et al (DOI 10.5194/gmd-12-2307-2019) all included both IEPOX and non-IEPOX SOA pathways along with detailed gas-phase isoprene chemistry into global models, and showed the relative contributions of each of the pathways. While the mechanism in this manuscript includes more detailed particle-phase chemistry than any of those studies, the chemistry of the SOA precursors in Bates & Jacob appears to be of comparable complexity. At the very least these previous studies should be mentioned and their mechanisms cursorily compared; most helpful to readers and to the field in general would be some discussion of how the model outcomes compare, e.g. the relative contributions from the various SOA-forming pathways.*

Response:

Thank you for your suggestion and we have included references to these studies. In our revised manuscript, we have included the introduction and discussion of these studies in the introduction section. The main and relevant model outcomes were briefly mentioned and compared.

L95-118: "To overcome these issues, a few recent studies have attempted to apply intermediate-size mechanisms in large-scale models which include the isoprene gas-phase oxidation scheme to certain extent of details (Stadtler et al., 2018; Bates and Jacob, 2019; Müller et al., 2019). This implementation allowed for simulations of the key gas-phase products such as ISOPOOH and

IEPOX, which turned out to be very similar to the MCM mechanism (Müller et al., 2019). The model investigated by Bates and Jacob (2019) estimated that the global production of isoprene SOA is about one-third from each of IEPOX, C5-NLV, and C5-LV. Nevertheless, not all the important SOA species and formation pathways were included in these mechanisms; gas-particle partitioning and particle-phase chemistry were not always considered and simplified parameterizations were still used in some of these models; systematic validation of these mechanisms against laboratory and field measurements was also lacking.

In this study, we developed a new condensed multiphase isoprene oxidation chemical mechanism adapted to the SAPRC structure (Carter, 2010). The new mechanism represents the isoprene chemistry with intermediate level of chemical details to include the major SOA species. It was also made flexible for the inclusion of new isoprene chemistry that is reported in laboratory, mechanistic, and field studies [e.g., (Wennberg et al., 2018; Vasquez et al., 2020; Mettke et al., 2022; Carlsson et al., 2023)]. Lastly, this mechanism is also implementable into regional or global air quality models to better represent isoprene chemistry and SOA formation. This mechanism was incorporated into a box model to simulate existing isoprene oxidation chamber experiments under various initial conditions (e.g., OH oxidation, NO3 oxidation, and different NOx levels, etc.). The key gas-phase products from all the pathways described above and SOA mass concentrations were compared with laboratory observations (where available) and other chemical mechanisms to evaluate the mechanism's performance. We also applied the new mechanism to model the 2013 Southern Oxidant and Aerosol Studies (SOAS) field campaign at the Centreville, AL site (Lee et al., 2016; H. Zhang et al., 2018), to elucidate the relative importance of the various pathways in SOA formation under real atmospheric conditions. To the best of our knowledge, this is the first time that a comprehensive molecular-level isoprene SOA mechanism is evaluated using field observations. Lastly, we also discuss the major uncertainties in current mechanistic understandings and the needed future research directions regarding isoprene SOA chemistry."

*Abstract: I know you don't want to sell yourselves short, but it would be helpful to highlight more the way that C5-LVN appear to be the outlier in how well the model works. While you mention that the model-measurement agreement breaks down under high-NO conditions, you don't mention the direction of the bias, and then you gloss over the sharp disagreement with ambient data as shown in Figure 5C. I think mentioning these would motivate readers by showing what isn't yet well-understood, since it's clear the mechanism is good at representing what we \*do\* understand!*

Response:

Thank you for your suggestion. We have revised our abstract as the following to include more details of the model results:

L20-29: "Our results show that SOA formation from most of the chamber experiments is reasonably reproduced using our mechanism except when the concentration ratios of initial nitric oxide to isoprene exceeds ~2, the formed SOA is significantly underpredicted. The SOAS simulations also reasonably agree with the measurements regarding the diurnal pattern and concentrations of different product categories while the total isoprene SOA remains

underestimated. The molecular compositions of the modelled SOA indicate that multifunctional low-volatility products contribute to isoprene SOA more significantly than previously thought, with a median mass contribution of ~57% to the total modelled isoprene SOA. However, this contribution is intricately intertwined with the IEPOX-derived SOA, posing challenges for their differentiation using bulk aerosol composition analysis (e.g., the aerosol mass spectrometer with positive matrix factorization). Furthermore, the SOA from these pathways may vary greatly, mainly dependent on the volatility estimation and treatment of particle-phase processes (i.e., photolysis and hydrolysis)."

*From Figure 1: which I realize is not meant to represent all the reactions in UCR-ISOP comprehensively, but it's all I have to go off -- it appears that many of the low volatility species are gas-phase "dead-ends", in that they lack loss pathways via reaction with OH. Is this the case, or were those reactions just left out of the figure? Either way, it would be helpful to describe whether such reactions exist in the mechanism and if so, how their rates and products were decided, since most such reactions have not been experimentally studied.*

Response:
Thank you for your query regarding Figure 1. Indeed, the figure was not intended to illustrate all the reactions within the UCR-ISOP mechanism comprehensively. It focuses on the principal pathways for clarity in presentation. The low volatility species can indeed react with OH radicals and are also subject to photolysis decay, and the reaction rates for these processes are taken either from the well-established Caltech mechanism or extrapolated from analogous reactions in MCMv3.3.1. To address your suggestion and provide clarity, we have updated the manuscript to include a description of these reactions. This additional information can now be found in Figure 1 caption and section 2.2 of the revised manuscript.

L78-80: "Simplified reaction scheme for isoprene oxidation by OH and $NO_3$. The major low-volatility species that may contribute to SOA formation are highlighted in dashed boxes. For simplicity, further reactions of the low-volatility species with OH and photolysis, as well as $RO_2$ + $RO_2$ reactions are not shown in the scheme, but they are included in the mechanism."

L156-157: "… and (4) the loss pathways of C5-LV and C5-NLV species via reactions with OH and photolysis, as suggested either by the Caltech mechanism or extrapolated from analogous reactions in MCM v3.3.1."

*L 203-204: Any estimation of how much this assumption of homogeneity might influence your results -- i.e., how much the model outcomes could differ is particles were allowed to adopt core-shell morphologies, since that has been estimated and studied previously (see e.g. DOI 10.1021/acsearthspacechem.1c00156)?*

Response:

Thank you for your insightful question. To address the potential impact of particle morphology on our model outcomes, we conducted additional simulations incorporating a core-shell structure. The organic and inorganic mass concentrations from AMS data are used to derive particle's core and shell thickness, with the assumption that the organic shell contains 10% of the aerosol-associated liquid water content and the inorganic core 90%. The products of $H_{org}$ (Henry's law coefficient in the organic layer) and $D_{org}$ (diffusion coefficient of IEPOX in the organic layer) are extrapolated from experimental results in Zhang et al. (2018). The comparison between modelled IEPOX-SOA using homogeneous vs. core-shell structures is shown below and added in the supplementary information. It can be seen that the modelled IEPOX-SOA concentration can be reduced by around 30% at daily peak when core-shell structure is assumed. We have added this description in the revised manuscript as the following:

L222-224: "Additional description of this process can be found in Text S1 in the Supplementary Material. In the model, the aerosols were assumed as a homogeneous mixture of organic and inorganic constituents. However, the influence of core-shell particle morphology on reactive uptake is simulated based on the method reported by Zhang et al. (2018) in sensitivity analysis."

In parallel with the influence of core-shell morphology on reactive uptake, we also considered non-ideal gas-particle partitioning in the sensitivity analysis for our SOAS simulations and added the relevant text in the revised manuscript:

L169-178: "In general, the condensation kinetics to particle is calculated as:

$$K_{cond} = \left( \frac{r_p}{D_g} + \frac{4}{\alpha\omega} \right)^{-1} \times \text{SA}, \quad (1)$$

where $K_{mt}$ is the condensation rate (s⁻¹), $r_p$ is the particle radius (cm) obtained from particle size measurements, $D_g$ is the gas-phase diffusivity (cm² s⁻¹), $\alpha$ represents the mass accommodation coefficient, $\omega$ is the molecular mean thermal velocity (cm s⁻¹), and SA is the aerosol surface area per volume (cm² cm⁻³). In ideal gas-particle partitioning assumptions, $\alpha = 1$ is used; while for non-ideal partitioning (e.g., in the presence of diffusion limitation), $\alpha$ in the range of 0.1 – 1 has been used (Saleh et al., 2013; Zhang et al., 2015). In Thornton et al. (2020), it was suggested that this range of $\alpha$ value has little impact on simulated isoprene SOA under the chamber experimental conditions. In this work, we will test the influence of $\alpha$ in the SOAS simulations in sensitivity analysis."

L497-503: "Furthermore, because the particle phase state is very important for gas-particle partitioning of low-volatility species and the reactive uptake of IEPOX (Zhang, Y. et al., 2018), the average O:C ratio (derived from the AMS measurements), organic mass to sulfate ratio and ambient RH were used to determine the particle phase and occurrence of phase separation behavior (Schmedding et al., 2020). Aerosols were found to be in liquid-like phase and internally mixed for most of time during the SOAS campaign at ground level (Fig. S9), suggesting that the usage of = 1 and homogeneous mixing of inorganic and organic species are valid. However, as described above, we still examined the influence of non-ideal partitioning and core-shell morphology on simulated isoprene SOA in sensitivity analyses."

L600-606: "However, a discernible underestimation of our modelled total isoprene SOA is present, when compared to the IEPOX-SOA factor (Fig. S15), suggesting additional SOA formation pathways from isoprene oxidation in the atmosphere that our mechanism does not include. Furthermore, the observed discrepancy between modelled results and actual measurements could be exacerbated if slow gas-particle partitioning and core-shell particle morphology are considered. In the sensitivity test shown in Fig. S18, such an adaptation in the model could lead to a further decrease in peak IEPOX-SOA estimates by ~ 40%, consistent with Zhang et al. (2018), but negligible change for non-IEPOX SOA due to slow particle diffusion and partitioning, consistent with Thornton et al. (2020)."

[Figure]

*L 214: Are other tertiary nitrates allowed to hydrolyze quickly with the Vasquez et al rate, or just 1,2-IHN? Following hydrolysis, are the products treated explicitly and allowed to repartition and react? (See e.g. DOI 10.1021/acs.est.1c04177).*

Response:

In the gas phase, we have limited the reactive uptake by particle liquid water exclusively to 1,2-IHN to form a diol (IDH). If future research suggests additional organonitrates following similar rapid hydrolysis, they can be readily added to the mechanism. In the particle phase, all low-volatility organic nitrates undergo hydrolysis with an average lifetime of 3 hours, through which the -$ONO_2$ group is converted to the –OH group (Pye et al., 2015). Post-hydrolysis, the products are indeed treated explicitly, allowing for their repartitioning and subsequent reactions. The related information can be found in section 2.4 and 2.5 in the manuscript:

L233-235: "In the case of 1,2-IHN, the reactive uptake product is expected to be a diol (IDH) via hydrolysis that is expected to quickly evaporate back to the gas phase. The reaction rate is calculated from LWC, $H_{aq}$, and the aqueous hydrolysis rate used in Vasquez et al. (2020)."

L255-257: "In the SOAS campaign simulations, because of the high relative humidity (RH) and LWC at the field site, we also apply hydrolysis reactions for the organic nitrate species in the simulated isoprene SOA. We assume that their average lifetime against hydrolysis is 3 hours, through which the $-ONO_2$ group is converted to the $-OH$ group (Pye et al., 2015)."

L 231: Particle-phase hydroperoxide photolysis has been mentioned, but how were gas-phase photolysis rates applied? Were all hydroperoxides assumed to photolyze there too? What about carbonyl nitrates? (See e.g. DOI 10.5194/acp-14-2497-2014).

Response:

We have determined gas-phase photolysis rates by referencing the established Caltech mechanism or extrapolating from similar reactions in the Master Chemical Mechanism (MCM v3.3.1). For instance, dihydroxy dihydroperoxides (IDHDP) are assigned a photolysis rate of twice that of J41, as J41 represents the photolysis rate for CH3OOH and thus serves as a proxy for the photolysis of similar organic peroxides. This rate was doubled for IDHDP to account for its two hydroperoxide functional groups. For other low-volatility hydroperoxides, photolysis rates are similarly scaled to J41. As for carbonyl nitrates, the rates are a composite of J53, J54, or J55 (for organic nitrates) and J22 (for carbonyl products, extrapolated from methyl ethyl ketone photolysis). We acknowledge that this approach is a simplified assumption, and more future studies need to be done to refine these estimations:

L156-157: "… and (4) the loss pathways of C5-LV and C5-NLV species via reactions with OH and photolysis, as suggested either by the Caltech mechanism or extrapolated from analogous reactions in MCM v3.3.1."

L 474: "were" should be "was"

L 489: "overpredict" should be "overpredicts"

Response:

Thank you for so detailed suggestions. We corrected these issues in the revised manuscript.

L 497: This potential involvement of cloud interactions as a loss pathway is a great idea! Is there any way to test of parameterize this, e.g. simply using a cloud presence flag from the measurement site to see if the loss correlates with the presence of clouds?

Response:

We are grateful for your positive feedback on considering cloud interactions as a potential loss pathway. Unfortunately, our current dataset from field measurements does not include cloud data, which makes the parameterization or correlation test impossible. We share your concern regarding this factor and think that it deserves a dedicated and thorough future study, possibly as a focal point itself, to understand the importance of cloud-chemical interactions in atmospheric science. But for now, it falls outside our current scope of investigation.

*L 569-577: The conclusions (a) that the widely used AMS-PMF factor for IEPOX-OA may in fact be convoluted with other isoprene SOA formation pathways, and (b) that total isoprene SOA remains drastically underestimated, both seem very important (particularly to people in this field that frequently come across the AMS-PMF analysis) and worth more highlighting. I would suggest that both of these merit a mention in the abstract and that some version of Figure S15 merits inclusion in the main manuscript. Does the AMS-PMF factor really include SOA from both LV pathways, as implied on L 575, or would the C5-NLV pathway be separable because the high-NOx products are from different chemical pathways (and presumably therefore have different temporal patterns)?*

Response:

We appreciate your input on our findings' significance. Your comment prompted us to further investigate the correlations separately, resulting in an $R^2$ of 0.82 for IEPOX-SOA with C5-LV and 0.35 for C5-NLV. This highlights a stronger convolution of IEPOX-SOA with C5-LV, likely due to shared precursors of ISOPOOH, whereas C5-NLV is more distinct and influenced by other factors like NOx conditions. Based on this analysis, we agree that the C5-NLV pathway appears more discernible within the AMS-PMF framework than C5-LV. We have emphasized these insights in the abstract and revised the following text in the main manuscript:

L22-29: "… The SOAS simulations also reasonably agree with the measurements regarding the diurnal pattern and concentrations of different product categories while the total isoprene SOA remains underestimated. The molecular compositions of the modelled SOA indicate that multifunctional low-volatility products contribute to isoprene SOA more significantly than previously thought, with a median mass contribution of ~57% to the total modelled isoprene SOA. However, this contribution is intricately intertwined with the IEPOX-derived SOA, posing challenges for their differentiation using bulk aerosol composition analysis (e.g., the aerosol mass spectrometer with positive matrix factorization). Furthermore, the SOA from these pathways may vary greatly, mainly dependent on the volatility estimation and treatment of particle-phase processes (i.e., photolysis and hydrolysis) …".

L590-594: "… In addition, we also compare the total simulated low-volatility SOA (C5-LV and C5-NLV) in correlation with the IEPOX-SOA with hourly resolution (Fig. S17). It turns out that the isoprene SOA from these two formation pathways correlated reasonably well (R2 = 0.80). In particular, calculated $R^2$ is 0.82 for IEPOX-SOA with C5-LV and 0.35 for C5-NLV, which highlights a stronger convolution of IEPOX-SOA with C5-LV than C5-NLV".

*L 577: "suggests" should be "suggesting"*

*L 592: "that of that from" ... I think either needs to be "that of" or "that from", but not both?*

Response:

Thank you for so detailed suggestions. We corrected these issues in the revised manuscript.

*L 594-595: This ratio of IHN to [ISOPOOH + IEPOX] doesn't seem at all to be an apples-to-apples comparison; the latter incorporates two generations of chemistry, and the lifetime of IEPOX with respect to OH oxidation is on the order of 5x longer than IHN. If the observed instantaneous ratio of IHN to [ISOPOOH + IEPOX] is 1:10, the ratio of the amount of carbon that goes through IHN to that which goes through the low-NOx, ISOPOOH+IEPOX pathway is probably closer to 1:2, given the difference in their lifetimes. While your point still stands that there's a surprisingly large amount to C5-NLV contributing to isoprene-derived SOA in a seemingly low-NOx environment, I would avoid characterizing that as IHN being 10x more potent on a per-carbon basis at SOA formation.*

Response:

We appreciate the depth of your analysis and agree that this is not an apple-to-apple comparison quantitatively. The goal of this is to approximately compare the efficiency of SOA formation in the low-NOx (ISOPOO + HO$_2$) vs. high-NOx (ISOPOO + NO) pathways. Under a low-NOx environment such as the southeastern U.S., it is surprising to us that C5-NLV contributes substantially to the modelled isoprene SOA. Thus, we traced it back to the precursors in a few steps. ISOPOOH + IEPOX and IHN are compared because (1) they are the key precursors for all the C5-LV and C5-NLV, respectively, and (2) they are quantified by measurements, providing high confidence in concentrations. From Fig. 6A-B to Fig. 6C-D, the ratio of ISOPNOO/ISOPOOHOO + IEPOXOO (~ 0.25) is higher than that of IHN/ISOPOOH+IEPOX (~ 0.1) is in fact partly due to the smaller IHN lifetime. This is consistent with our analysis. In the revised manuscript, we have incorporated the aspect of chemical lifetime into our discussion as follows:

L623-628: "The strong contrast suggests that the IHN oxidation pathway is much more efficient in producing LV SOA than the ISOPOOH/IEPOX + OH pathway. We suggest that this results from several effects. First, the OH oxidation of ISOPOOH and IEPOX produce C5-LV RO2 (i.e., ISOPOOHOO and IEPOXOO) at smaller branching ratios than C5-NLV RO2 (ISOPNOO) production from IHN + OH (Wennberg et al., 2018), and IEPOX has a lifetime of ~5 times longer than IHN against OH. These kinetic differences lead to a reduced difference between the production rates of the C5-LV RO2 and the C5-NLV RO2 to a factor of ~4 (Fig. 6C–D)."

However, we respectfully disagree with the reviewer's last point on the estimated ratio of carbon goes through ISOPOOH+IEPOX vs. IHN, because the ratio in lifetime does not always factor into

the concentration by the same scale. To demonstrate this point, we compare modelled production rates of ISOPOOH vs. IHN (both are first-generation products from ISOPOO + HO2 vs. ISOPOO + NO). We found that the ratio in the production rate of IHN/ISOPOOH (~0.12) is only slightly higher than IHN/ISOPOOH+IEPOX shown in Fig. 6.

[Figure]

*L 596: "is resulted" should be "results"*

Response:

Thank you for the suggestion. We corrected this issue in the revised manuscript.

*Reviewer #2:*

*This paper introduces a new isoprene SOA multiphase chemical mechanism and provides detailed analysis of the efficacy of the mechanism against both laboratory and field studies under a variety of different conditions. The results are very detailed and thorough with much discussion into both what the model does well and its limitations. However, I think a lack of detail on the specifics of the mechanism led to confusion about what the goal of introducing this mechanism is and how it is advancing the science beyond the currently existing isoprene SOA mechanisms. Overall the work fits well within the scope of ACP, the results are thorough, and the limitations are clear and well-described. I would recommend publication in ACP once a few comments are addressed.*

1. *How were the reactions and species that were included in the model determined? Was the larger MCM reduced or were selected species/reactions added to SAPRC07? Either way additional description on how this was done is warranted.*

Response:

We thank the reviewer for the comment. Our mechanism was developed on top of the SAPRC07 mechanism; thus, we have maintained the naming for species carried over from SAPRC07tic while adopting the Caltech mechanism's naming convention for isomers sharing functional groups. For instance, isomers featuring two hydroxy and two hydroperoxide groups are collectively termed IDHDP. However, specific isomers such as ISOPOHOO, HPALD, ISOPOOH, IEPOX, and IHN are individually detailed due to their established distinct reactivities, as per Wennberg et al., 2018. This balance of lumping and separating isomers enhances the accuracy of our product distribution predictions.

We have revised the following paragraph to better explain our naming strategy in our mechanism:

L137-151: "In UCR-ISOP, for the species that were already included in the SAPRC07tic mechanism, we have preserved their nomenclature. For other species, the naming convention is the same as in the Caltech mechanism, which lumps the isomers with the same functional groups to one compound. It is to be noted that our new mechanism includes many multifunctional C5 species which were not included in the SAPRC07 mechanism but pivotal as SOA precursors. For example, isomers with two hydroxy (−OH) and two hydroperoxide (−OOH) groups are now represented by a single species, IDHDP. Thus, each of the low-volatility species shown in Fig. 1 are described as an individual compound that could represent the sum of several isomers. On the contrary, certain isomers are individually represented (with some extent of lumping in certain cases) for several major species, including the isoprene hydroxyl peroxy radicals (ISOPOHOO, 2 isomers), hydroperoxyl aldehydes (HPALD, 2 isomers), ISOPOOH (3 isomers), IEPOX (2 isomers), and isoprene hydroxynitrates (IHN, 3 isomers). These species have been extensively studied in the literature and distinct reactivities and reaction products have been reported (Wennberg et al., 2018). Maintaining some of the lumped isomers for these species permits more accurate representations of their further product distributions. All the abbreviated names in the UCR-ISOP mechanism are described in Table S1 in the Supplementary Material. Compared to SAPRC07tic (38 species and 124 reactions), UCR-ISOP adds 39 additional species and 118

additional gas-phase reactions. In comparison, the MCM mechanism has 610 species and 1974 reactions (related to isoprene); the Caltech mechanism has 155 species and 429 reactions."

2. *Is any comparison possible with machine learning-based reduced mechanisms (e.g. Wiser et al)? If direct comparison is not possible, I think a discussion of the pros/cons of this reduced model compared to machine learning-based mechanisms would be useful.*

Response:

We appreciate your suggestion regarding the comparison with machine learning-based reduced mechanisms. We are aware of the work by Wiser et al. and the progress of implementing machine learning to reduce chemical mechanisms for cost-efficient large-scale modeling. At present, our study focuses on including more species and reactions that are key to isoprene SOA formation and evaluate this implementation to gain some detailed mechanistic understanding of isoprene SOA formation chemistry. For this purpose, we compare our mechanism with mechanisms like the Caltech mechanism, MCM v3.3.1, and MCM-UW, which represent the more comprehensive isoprene mechanisms currently in use. While we recognize the value of machine learning-based models, it appears to us that the goals are in opposite directions and including such a discussion would deviate our focus. Therefore, we decided to concentrate on these established mechanisms to maintain the focus and conciseness of our paper. Your understanding is appreciated. However, we do acknowledge the potential interest in such comparisons and suggest this could be a valuable direction for future research.

3. *More discussion would be helpful on how the mechanisms discussed in this work differ from each other beyond just the size of the mechanism. Since all the mechanisms seem to perform similarly in their ability to predict isoprene SOA, it is important for the authors to indicate what this new mechanism is adding beyond those that already exist.*

Response:

Thank you for your comments. It should be noted that not all mechanisms equally predict isoprene SOA. For instance, the SAPRC mechanism does not account for SOA from low-volatility pathways. During nighttime oxidation, MCM v3.3.1 and MCM-UW significantly underestimate SOA formation, while the Caltech mechanism does not accurately capture SOA variations as our new mechanism does (see Fig.S6). These mechanisms also do not include dimer formation pathways, which has been considered an important pathway for isoprene SOA formation. In order to indicate what this new mechanism is adding beyond those that already exist, we have revised the following paragraph in the manuscript:

L151-159: "In addition, our mechanism also incorporates many up-to-date theoretical or experimental findings on isoprene oxidation and SOA formation, including (1) temperature and pressure dependence of organic nitrate yield from peroxy radical (RO2) + NO reactions as

suggested by the Caltech mechanism; (2) isomerization reactions for the major RO2 based on recent studies (D'ambro et al., 2017; Wennberg et al., 2018; Vereecken et al., 2021; Mettke et al., 2023); (3) dimer formation from several RO2 + RO2 reactions that were supported by prior chamber experiments (Ng et al., 2008; Mettke et al., 2023); and (4) the loss pathways of C5-LV and C5-NLV species via reactions with OH and photolysis, as suggested by either the Caltech mechanism or extrapolated from analogous reactions in MCM v3.3.1. For the isoprene + NO3 reactions, the new FZJ mechanism proposed by recent studies was also incorporated to some extent as discussed later (Vereecken et al., 2021; Tsiligiannis et al., 2022; Carlsson et al., 2023)."

Additional details of these differences were also discussed in section 3.1.2.

4. *In particular, the comparison in Fig 2 seems to show that there is very little improvement in the UCR model from the SAPRC07 mechanism. In light of that, what is the additional chemistry representing better?*

Response:

Our UCR mechanism enhances the SAPRC07 by integrating multifunctional C5 species that are pivotal as SOA precursors. These are absent in SAPRC07, meaning that SOA formation from this pathway is notably missing in the SAPRC07 mechanism. Moreover, we incorporate recent research on isomerization reactions of major RO2 species, dimer formation via RO2+RO2, and updated NO3 oxidation processes during the nighttime. These refinements are crucial for a more accurate representation of isoprene oxidation and SOA formation. While Figure 2 is confined to VOC-O3-NOx comparisons and does not depict the full extent of these enhancements, our supplementary materials, like Fig.S6, provide a complementary view of our mechanism's superior performance in SOA simulation. The main goal of comparing VOC-O3-NOx model performance is to ensure that with the additions of SOA precursors, the predictions of VOC, O3, and NOx are not negatively affected. This point is clarified in the revised manuscript:

L291-294: "The performances of all the compared mechanisms are in general similar to each other. These results support that with the new additions of species and pathways key to isoprene SOA formation on top of the SAPRC07 mechanism as well as reduction of the more explicit mechanisms into UCR-ISOP, the capability to accurately predict isoprene, $O_3$, and $NO_x$ are not negatively affected"

5. *Does the mechanism include gas-phase RO2+RO2 reactions to form dimers? This is mentioned as a possible pathway during the results, but it was not introduced in the methods, which led to some confusion for me.*

Response:

Thank you for your query regarding the inclusion of RO2+RO2 dimer formation in our mechanism. This process is indeed incorporated and is mentioned in Section 2.2 as part of the overall mechanism overview. A more detailed discussion is presented in Section 3.1.2, where we delve into specific reactions pertinent to the modeling results. We opted for this structure to maintain a clear and accessible introduction before exploring the complexities within the results section.

L155-156: "… (3) dimer formation from several RO2 + RO2 reactions that were supported by prior chamber experiments (Ng et al., 2008; Mettke et al., 2023);…"

L323-326: "In addition, we consider the rapid ISOPOOHOO self-reaction to form the corresponding carbonyl (C5H10O6), alcohol (C5H12O6), and dimer (C10H22O10), suggested by Mettke et al. (2023), with a rate coefficient of $1 \times 10^{-11}$ cm$^3$ molecule$^{-1}$s$^{-1}$. This dimer formation pathway could partly explain the slightly lower C5-LV simulation using the UCR-ISOP mechanism under high concentrations (Fig. 3E)."

L374-380: "In a previous study, Kwan et al. (2012) proposed dimer formation from NISOPO2 + NISOPO2 with a branching ratio of 3–4% based on gas-phase measurements. However, Ng et al. (2008) observed a substantial amount of dimers in the SOA from the same experiments, suggesting that the actual dimer formation branching ratio could be much higher given their very low volatility. In UCR-ISOP, we assume this branching ratio to be 10%, which leads to a good agreement with the SOA simulation (see next section). This dimer formation pathway from NISOPO2 + NISOPO2 could also partly explain the slightly lower C5-NLV simulation using the UCR-ISOP mechanism under high concentrations (Fig. 3F)."

6. *The resistor model for uptake coefficient assumes no particle-phase diffusion to uptake, however should IEPOX-SOA form a viscous organic shell around an acidic core, this may affect the uptake. While this would not help the modeled underestimation of IEPOX-SOA, do the authors expect this to impact the results in this work at all?*

Response:

Thank you for your insightful question. To address the potential impact of particle morphology on our model outcomes, we conducted additional simulations incorporating a core-shell structure. The organic and inorganic mass concentrations from AMS data are used to derive particle's core and shell thickness, with the assumption that the organic shell contains 10% of the aerosol-associated liquid water content and the inorganic core 90%. The products of $H_{org}$ (Henry's law coefficient in the organic layer) and $D_{org}$ (diffusion coefficient of IEPOX in the organic layer) are extrapolated from experimental results in Zhang et al. (2018). The comparison between modelled IEPOX-SOA using homogeneous vs. core-shell structures is shown below and added in the supplementary information. It can be seen that the modelled IEPOX-SOA concentration can be reduced by around 30% at daily peak when core-shell structure is assumed. We have added this description in the revised manuscript as the following:

L222-224: "Additional description of this process can be found in Text S1 in the Supplementary Material. In the model, the aerosols were assumed as a homogeneous mixture of organic and inorganic constituents. However, the influence of core-shell particle morphology on reactive uptake is simulated based on the method reported by Zhang et al. (2018) in sensitivity analysis."

In parallel with the influence of core-shell morphology on reactive uptake, we also considered non-ideal gas-particle partitioning in the sensitivity analysis for our SOAS simulations and added the relevant text in the revised manuscript:

L169-178: "In general, the condensation kinetics to particle is calculated as:

$$K_{cond} = \left(\frac{r_p}{D_g} + \frac{4}{\alpha\omega}\right)^{-1} \times SA, \quad (1)$$

where $K_{mt}$ is the condensation rate (s$^{-1}$), $r_p$ is the particle radius (cm) obtained from particle size measurements, $D_g$ is the gas-phase diffusivity (cm$^2$ s$^{-1}$), $\alpha$ represents the mass accommodation coefficient, $\omega$ is the molecular mean thermal velocity (cm s$^{-1}$), and SA is the aerosol surface area per volume (cm$^2$ cm$^{-3}$). In ideal gas-particle partitioning assumptions, $\alpha = 1$ is used; while for non-ideal partitioning (e.g., in the presence of diffusion limitation), $\alpha$ in the range of $0.1 - 1$ has been used (Saleh et al., 2013; Zhang et al., 2015). In Thornton et al. (2020), it was suggested that this range of $\alpha$ value has little impact on simulated isoprene SOA under the chamber experimental conditions. In this work, we will test the influence of $\alpha$ in the SOAS simulations in sensitivity analysis."

L497-503: "Furthermore, because the particle phase state is very important for gas-particle partitioning of low-volatility species and the reactive uptake of IEPOX (Zhang, Y. et al., 2018), the average O:C ratio (derived from the AMS measurements), organic mass to sulfate ratio and ambient RH were used to determine the particle phase and occurrence of phase separation behavior (Schmedding et al., 2020). Aerosols were found to be in liquid-like phase and internally mixed for most of time during the SOAS campaign at ground level (Fig. S9), suggesting that the usage of = 1 and homogeneous mixing of inorganic and organic species are valid. However, as described above, we still examined the influence of non-ideal partitioning and core-shell morphology on simulated isoprene SOA in sensitivity analyses."

L600-606: "However, a discernible underestimation of our modelled total isoprene SOA is present, when compared to the IEPOX-SOA factor (Fig. S15), suggesting additional SOA formation pathways from isoprene oxidation in the atmosphere that our mechanism does not include. Furthermore, the observed discrepancy between modelled results and actual measurements could be exacerbated if slow gas-particle partitioning and core-shell particle morphology are considered. In the sensitivity test shown in Fig. S18, such an adaptation in the model could lead to a further decrease in peak IEPOX-SOA estimates by $\sim 40\%$, consistent with Zhang et al. (2018), but negligible change for non-IEPOX SOA due to slow particle diffusion and partitioning, consistent with Thornton et al. (2020)."

[Figure]

7. *In Fig 2, what was considered an "outlier"? Beyond the discussed issue reproducing high NOx experiments, was there any consistency in these outliers that were not represented well?*

Response:
Points are drawn as outliers if they are larger than Q3+W*(Q3-Q1) or smaller than Q1-W*(Q3-Q1), where Q1 and Q3 are the 25th and 75th percentiles, respectively. The default value of W is 1.5, which corresponds to approximately +/- 2.7 sigma and 99.3 coverage if the data are normally distributed.

Upon examining data points that poorly predicted ISOP/NO/O3 levels, we noted a temporal clustering of these outliers, suggesting possible issues with the experimental data from specific periods. For instance, the experiment labeled 20120608N was shown below and marked as an outlier. It can be seen that the NO readings did not show a consistent decline, potentially due to sensor crosstalk as it alternated between two chambers. Although the exact nature of these discrepancies is uncertain, we chose to include these data points in our analysis to maintain transparency and completeness of the dataset.

[Figure]

Minor comment

1. *Fig 4. Please check the +/-50% lines as one appears to be 1.5:1 and one appears to be 1:2. I assume these were meant to be the same ratio.*

   Response:
   Thank you for your attention to Figure 4. The lines you've referred to are indeed intended to depict a ±50% variation. The upper line represents a 50% increase, formulated as $y = (1 + 0.50) * x$, and the lower line represents a 50% decrease, expressed as $y = (1 - 0.50) * x$. This is meant to illustrate the range of variability around the line of unity (x=y) for clarity in our data representation.

*References*

*Wiser, F., Place, B. K., Sen, S., Pye, H. O. T., Yang, B., Westervelt, D. M., Henze, D. K., Fiore, A. M., and McNeill, V. F.: AMORE-Isoprene v1.0: a new reduced mechanism for gas-phase isoprene oxidation, Geosci. Model Dev., 16, 1801–1821, https://doi.org/10.5194/gmd-16-1801-2023, 2023.*

Kwan, A., Chan, A., Ng, N., Kjaergaard, H., Seinfeld, J., and Wennberg, P.: Peroxy radical chemistry and OH radical production during the NO 3-initiated oxidation of isoprene, Atmospheric Chemistry and Physics, 12, 7499-7515, 2012.

Lee, B. H., Mohr, C., Lopez-Hilfiker, F. D., Lutz, A., Hallquist, M., Lee, L., Romer, P., Cohen, R. C., Iyer, S., Kurtén, T., Hu, W., Day, D. A., Campuzano-Jost, P., Jimenez, J. L., Xu, L., Ng, N. L., Guo, H., Weber, R. J., Wild, R. J., Brown, S. S., Koss, A., de Gouw, J., Olson, K., Goldstein, A. H., Seco, R., Kim, S., McAvey, K., Shepson, P. B., Starn, T., Baumann, K., Edgerton, E. S., Liu, J., Shilling, J. E., Miller, D. O., Brune, W., Schobesberger, S., D'Ambro, E. L., and Thornton, J. A.: Highly functionalized organic nitrates in the southeast United States: Contribution to secondary organic aerosol and reactive nitrogen budgets, Proc. Natl. Acad. Sci. U.S.A., 113, 1516-1521, 2016.

Mettke, P., Brüggemann, M., Mutzel, A., Gräfe, R., and Herrmann, H.: Secondary Organic Aerosol (SOA) through Uptake of Isoprene Hydroxy Hydroperoxides (ISOPOOH) and its Oxidation Products, ACS Earth and Space Chemistry, 7, 1025-1037, 10.1021/acsearthspacechem.2c00385, 2023.

Ng, N. L., Kwan, A. J., Surratt, J. D., Chan, A. W. H., Chhabra, P. S., Sorooshian, A., Pye, H. O. T., Crounse, J. D., Wennberg, P. O., Flagan, R. C., and Seinfeld, J. H.: Secondary organic aerosol (SOA) formation from reaction of isoprene with nitrate radicals (NO$_3$), Atmos. Chem. Phys., 8, 4117-4140, 10.5194/acp-8-4117-2008, 2008.

Pye, H. O. T., Luecken, D. J., Xu, L., Boyd, C. M., Ng, N. L., Baker, K. R., Ayres, B. R., Bash, J. O., Baumann, K., Carter, W. P. L., Edgerton, E., Fry, J. L., Hutzell, W. T., Schwede, D. B., and Shepson, P. B.: Modeling the current and future roles of particulate organic nitrates in the Southeastern United States, Environ. Sci. Technol., 49, 14195-14203, 10.1021/acs.est.5b03738, 2015.

Saleh, R., Donahue, N. M., and Robinson, A. L.: Time Scales for Gas-Particle Partitioning Equilibration of Secondary Organic Aerosol Formed from Alpha-Pinene Ozonolysis, Environmental Science & Technology, 47, 5588-5594, 10.1021/es400078d, 2013.

Zhang, H., Worton, D. R., Shen, S., Nah, T., Isaacman-VanWertz, G., Wilson, K. R., and Goldstein, A. H.: Fundamental Time Scales Governing Organic Aerosol Multiphase Partitioning and Oxidative Aging, Environmental Science & Technology, 49, 9768-9777, 10.1021/acs.est.5b02115, 2015.

Zhang, Y., Chen, Y., Lambe, A. T., Olson, N. E., Lei, Z., Craig, R. L., Zhang, Z., Gold, A., Onasch, T. B., Jayne, J. T., Worsnop, D. R., Gaston, C. J., Thornton, J. A., Vizuete, W., Ault, A. P., and Surratt, J. D.: Effect of the Aerosol-Phase State on Secondary Organic Aerosol Formation from the Reactive Uptake of Isoprene-Derived Epoxydiols (IEPOX), Environ. Sci. Technol. Lett., 5, 167-174, 10.1021/acs.estlett.8b00044, 2018.